# NGF-TrkA signaling dictates neural ingrowth and aberrant osteochondral differentiation after soft tissue trauma

Seungyong Lee [1,8], Charles Hwang [2,8], Simone Marini[3], Robert J. Tower [4], Qizhi Qin[1], Stefano Negri [1,5], Chase A. Pagani[2], Yuxiao Sun[2], David M. Stepien[2], Michael Sorkin[2], Carrie A. Kubiak[2], Noelle D. Visser[2], Carolyn A. Meyers[1], Yiyun Wang [1], Husain A. Rasheed [2], Jiajia Xu[1], Sarah Miller[1], Amanda K. Huber[2], Liliana Minichiello [6], Paul S. Cederna[2], Stephen W. P. Kemp[2], Thomas L. Clemens[4,7], Aaron W. James [1✉] & Benjamin Levi[2✉]

Pain is a central feature of soft tissue trauma, which under certain contexts, results in aberrant osteochondral differentiation of tissue-specific stem cells. Here, the role of sensory nerve fibers in this abnormal cell fate decision is investigated using a severe extremity injury model in mice. Soft tissue trauma results in NGF (Nerve growth factor) expression, particularly within perivascular cell types. Consequently, NGF-responsive axonal invasion occurs which precedes osteocartilaginous differentiation. Surgical denervation impedes axonal ingrowth, with significant delays in cartilage and bone formation. Likewise, either deletion of *Ngf* or two complementary methods to inhibit its receptor TrkA (Tropomyosin receptor kinase A) lead to similar delays in axonal invasion and osteochondral differentiation. Mechanistically, single-cell sequencing suggests a shift from TGFβ to FGF signaling activation among pre-chondrogenic cells after denervation. Finally, analysis of human pathologic specimens and databases confirms the relevance of NGF-TrkA signaling in human disease. In sum, NGF-mediated TrkA-expressing axonal ingrowth drives abnormal osteochondral differentiation after soft tissue trauma. NGF-TrkA signaling inhibition may have dual therapeutic use in soft tissue trauma, both as an analgesic and negative regulator of aberrant stem cell differentiation.

[1] Department of Pathology, Johns Hopkins University, Baltimore, MD, USA. [2] Department of Surgery, Center for Organogenesis and Trauma, University of Texas, Southwestern, TX, USA. [3] Department of Epidemiology, University of Florida, Gainesville, FL, USA. [4] Department of Orthopaedics, Johns Hopkins University, Baltimore, MD, USA. [5] Department of Orthopaedics and Traumatology, University of Verona, Verona, Italy. [6] Department of Pharmacology, Oxford University, Oxford, UK. [7] Baltimore Veterans Administration Medical Center, Baltimore, MD, USA. [8] These authors contributed equally: Seungyong Lee, Charles Hwang. ✉email: awjames@jhmi.edu; Benjamin.Levi@utsouthwestern.edu

Soft tissue trauma such as musculoskeletal polytrauma, central nervous system trauma, or extensive burn injury causes the pathologic formation of cartilage and bone outside the skeleton, termed heterotopic ossification (HO)[1–10]. Trauma-induced HO can be conceived of as a tissue repair process gone awry, with abnormal differentiation of predominantly local mesenchymal progenitor cells, which inappropriately form bone via either endochondral or intramembranous ossification[11]. In its more severe manifestations, HO can induce chronic pain, impair wound healing, increase medical care utilization, lead to disability, and reduce the quality of life[11,12]. Treatment options for primary or recurrent HO are sparse[13], and medical therapies for HO are essentially limited to non-steroidal anti-inflammatory drugs and low-dose radiation[11]. Pain precedes HO in sporadic forms[14–16], and even rare genetic forms[17–19], which is often out of proportion to clinical examination. Given these clinical observations, we hypothesized that sensory innervation would play a pathoetiogenic role in trauma-induced HO.

A central mediator of bone pain is the nerve growth factor (NGF), which transmits nociceptive signals either by directly activating tropomyosin receptor kinase A (TrkA)-expressing nociceptive fibers or through indirect mechanisms, enhancing the response of other pain pathways[20,21]. Following activation, NGF–receptor complexes are internalized by endocytosis and form specialized signaling vesicles that undergo long-distance, retrograde transport from the distal axon to the cell body via the signaling endosome[22]. In the developing or injured mouse skeleton, TrkA-expressing sensory neurons invade the long bone perichondrium/periosteum at sites of NGF expression, where they are required for subsequent expansion and differentiation of skeletal progenitors accompanied by vascular assembly[23,24]. In lower vertebrates, an analogous paradigm for peripheral innervation appears to operate during tissue regeneration. For example, the regeneration of limbs of certain starfish and amphibians, and healing of fish scales and barbells all depend on neural inputs[25–27]. More recent studies show that mouse digit tip regeneration may likewise represent a peripheral nerve-dependent event[28–30]. Several preclinical studies in bone morphogenetic protein 2 (BMP2)-induced HO implicate peripheral nerves as a potential cellular contributor to HO formation[31–33]. Yet, a fundamental understanding of how peripheral innervation modulates abnormal mesenchymal stem cell fate decisions after trauma is lacking.

Here, we sought to define the regulatory role of neural inputs in a previously characterized trauma-induced HO model of aberrant cell fate[34–36]. During HO disease progression, NGF was expressed in several cell types, particularly vascular smooth muscle cells (SMCs) and pericytes, driving progressive sensory and sympathetic axonal invasion of the HO site. Surgical denervation impeded axonal ingrowth, abrogating endochondral bone formation within the injured soft tissue. Similarly, two transgenic animal models to interrupt NGF-TrkA signaling inhibited osteocartilaginous differentiation and blunted HO evolution. A clinically relevant small-molecule inhibitor of TrkA likewise abolished this aberrant cell fate differentiation mitigating HO formation. To explain these observations, single-cell sequencing (scSeq) suggested a shift from transforming growth factor-β (TGFβ) to fibroblast growth factor (FGF) signaling activation among pre-chondrogenic cells following the loss of neural inputs. Finally, analyses of human pathologic specimens and databases confirmed the relevance of NGF-TrkA signaling in human disease. In sum, NGF-responsive, TrkA-expressing peripheral neurons positively regulate trauma-induced HO. These findings suggest that NGF-TrkA signaling inhibition may have dual therapeutic use in soft tissue trauma, both as an analgesic and a negative regulator of aberrant stem cell differentiation.

## Results

**Axonal invasion accompanies aberrant cell fate seen in post-traumatic HO.** The temporospatial patterning of peripheral nerves during the course of heterotopic bone formation was first examined in a previously validated mouse model of trauma-induced HO[34–36]. Here, Achilles tenotomy coupled with a dorsal burn injury to incite systemic inflammation results in pathologic endochondral ossification of the tenotomy site over a defined 9-week period. Sagittal cross sections of mice hindlimb were examined, focusing on the distal tenotomy site (Fig. 1a). As we have previously reported, a phasic disease process ensues after injury, which recapitulates human disease, including a fibroproliferative phase (1 week) (Fig. 1b, c), cartilaginous phase (3 weeks) (Fig. 1d, e), and osseous phase (9 weeks) (Fig. 1f, g).

Nerve fibers within the HO site were first examined, using the pan-neuronal marker Beta III Tubulin (TUBB3) on sagittal sections of the distal tenotomy site (Fig. 1h–k). Inconspicuous TUBB3⁺ nerve fibers were found primarily within the peritenon of the uninjured Achilles tendon (Fig. 1h). A gradual increase in TUBB3⁺ nerve fibers was observed to invade the distal tenotomy site at 1, 3, and 9 weeks post injury (Fig. 1i–k). Quantification of TUBB3⁺ nerve fiber density confirmed these findings, demonstrating a mean 315%, 1128%, and 1650% increase in nerve density over baseline at 1, 3, and 9 weeks, respectively (Fig. 1l). Nerve fiber type was next characterized using immunohistochemical detection of either the sensory nerve marker calcitonin gene-related peptide (CGRP) or the sympathetic marker tyrosine hydroxylase (TH) (Fig. 1m–r). Inconspicuous CGRP⁺ nerve fibers were present in the peritenon at baseline (Fig. 1m), and had conspicuously increased at the outskirts of the injured tendon at the study endpoint (Fig. 1n). This reflected a 569% increase in CGRP⁺ nerve density (Fig. 1o). Similar patterns were observed among TH⁺ sympathetic fibers, which were less frequent within the uninjured peritenon (Fig. 1p), but conspicuously increased at the tenotomy site over 9 weeks (Fig. 1q). This reflected a 572% increase in TH⁺ nerve density over the study period (Fig. 1r). Thus, progressive axonal invasion accompanies post-traumatic heterotopic bone, composed of both peptidergic and sympathetic autonomic fibers.

**Proximal neurectomy abrogates post-traumatic aberrant cell fate.** In order to interrogate the requirement for axonal invasion, HO induction was performed in the context of surgical denervation. Here, HO induction was performed with or without ipsilateral sciatic nerve transection proximal to the trifurcation of the nerve (Fig. 2a). von Frey testing confirmed mechanical hypoalgesia within denervated animals (Supplementary Fig. S1a, 21.2% delay in paw withdrawal). As expected, among sham-operated conditions, TUBB3⁺ nerve fibers again showed progressive invasion of the HO site at 1, 3, and 9 weeks post injury in relation to 0-week control. In comparison, this progressive ingrowth was not seen among denervated HO sites (Fig. 2b–h). Quantitative histologic analysis was next performed using TUBB3-immunostained sections (Fig. 2i). This was quantified, demonstrating a progressive increase in overall TUBB3 immunoreactivity across the study period (303%, 1224%, and 1286% increase at 1, 3, and 9 weeks, respectively, in relation to control), while no significant increase in nerve fiber frequency was observed within denervated animals across this time course (Fig. 2i, 79% reduction at 9 weeks in comparison to sham-operated animals). These findings were confirmed using immunohistochemical identification of CGRP⁺ sensory fibers and TH⁺ sympathetic fibers within the HO site (Fig. 2j–o). CGRP immunostaining and quantitative analysis of the HO site demonstrated complete inhibition of CGRP⁺ fiber ingrowth (Fig. 2j–l, 66%

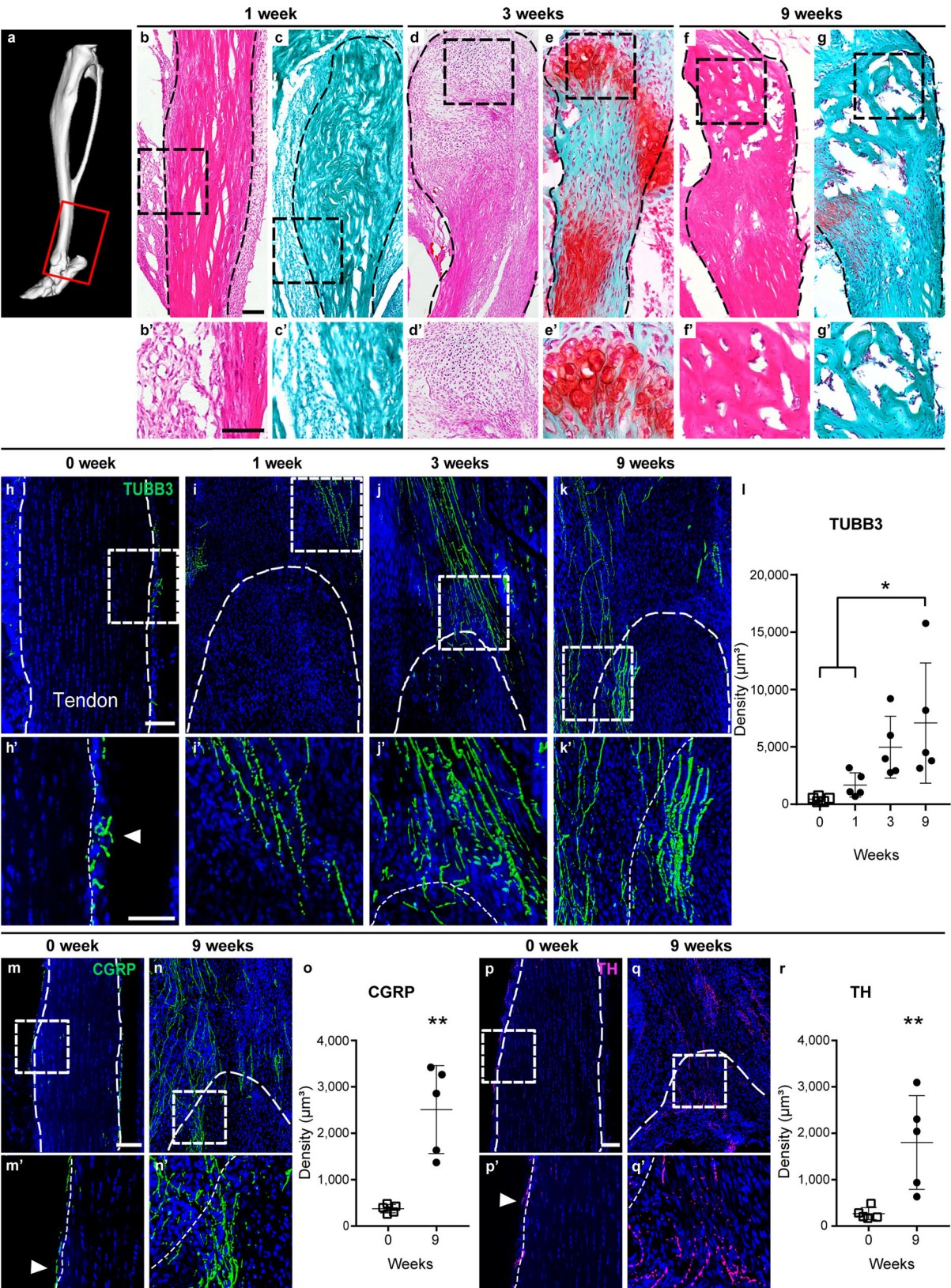

reduction in CGRP immunostaining). A concordant and complete abrogation of TH$^+$ fiber ingrowth was also observed, using TH immunohistochemistry and quantitative analysis (Fig. 2m–o, 68% reduction in TH immunostaining).

The effects of denervation on cartilage and bone formation were next assessed. Sagittal sections of the injured tendon were immunostained with SOX9 and stained with Safranin O to detect cartilage within the developing HO site (Fig. 2p–v). At 1 week post injury, increased SOX9 immunoreactivity was found among sham-operated sites, but not denervated injury sites (Fig. 2p, q). At 3 weeks, sham-operated animals showed an accumulation of stained cartilage glycosaminoglycans, but not in neurectomized

**Fig. 1 Progressive axonal invasion accompanies post-traumatic heterotopic ossification (HO). a** μCT reconstruction demonstrating location of analyses at the distal Achilles tendon. Red box represents the region of interest. **b–g** Illustration of endochondral ossification after HO induction, within the distal tenotomy site among mice subjected to combination tenotomy plus dorsal burn (burn/tenotomy). Dashed black line indicates the margin of the Achilles tendon. Dashed black box shows the high magnification images in panels (**b′–g′**). The progression of heterotopic ossification (HO) via an endochondral process, as visualized using sagittal sections of the distal tenotomy site. Tile scans appear above, while high magnification images are below. **b, c** Fibroproliferative phase at 1 week post injury, as shown by **b** hematoxylin and eosin (H&E) and (**c**) Safranin O-Fast Green (SO/FG) staining. **d, e** Cartilaginous phase at 3 weeks post injury, as shown by **d** H&E and **e** SO/FG staining. **f, g** Osseous phase at 9 weeks post injury, as shown by **f** H&E and **g** SO/FG staining. **h–k** Pan-neuronal Beta III Tubulin (TUBB3) immunohistochemical stains within representative sagittal sections of the distal tenotomy site, shown at 1, 3, and 9 weeks post injury in relation to uninjured control (0 week). **l** Quantification of TUBB3$^+$ nerve density within the injury site over the time course of HO formation (uninjured versus 9 weeks, $*p = 0.017$; 1 week versus 9 weeks, $*p = 0.049$). **m, n** Immunohistochemical staining for the peptidergic fiber marker calcitonin gene-related peptide (CGRP) at 9 weeks post injury in relation to uninjured control. **o** Quantification of CGRP$^+$ nerve density ($**p = 0.001$). **p, q** Immunohistochemical staining for the sympathetic fiber marker tyrosine hydroxylase (TH) at 9 weeks post injury in relation to uninjured control. **r** Quantification of TH$^+$ nerve density ($**p = 0.009$). Scale bars: 100 μm. Tile scans are presented above and dashed white lines indicate the margins of the Achilles tendon. Dashed white boxes represent the high magnification images shown below. White arrowheads show nerve fibers in the uninjured peritenon. For all graphs, each dot represents a single analyzed animal, with mean ± 1 SD indicated. $N = 5$ animals per analysis, per timepoint. Statistical analysis was performed using a two-sided one-way ANOVA with post hoc Tukey's test (**l**) or two-way Student's $t$ test (**o, r**). Source data are provided as a Source Data file.

animals (Fig. 2r, s). At 9 weeks, frank bone formation was observed within sham-operated control tissues, while cartilage was now observed among denervated hindlimbs (Fig. 2t, u). These findings of delayed endochondral ossification were quantified using histomorphometric analysis of serial sections to delineate cartilaginous area (C.Ar) within the injured site (Fig. 2v). Under sham-operated conditions, cartilaginous tissue areas were prominent at 3 weeks and barely detected at 9 weeks as endochondral ossification progressed. In contrast, cartilaginous tissue was much reduced at 3 weeks (Fig. 2v, 66% reduction in C. Ar in comparison to sham-operated animals) in denervated specimens. Residual cartilage was confirmed at 9 weeks among denervated animals. This reduction in endochondral ossification was confirmed using microcomputed tomography (μCT), assessed at the study endpoint (Fig. 2w, x). Across all samples, extraskeletal ossification was apparent in sham-operated specimens, but not in denervated specimens, which was confirmed by a significant reduction in bone volume (BV) by μCT analysis (Fig. 2y, 97% reduction among denervated animals). Essentially complete abrogation of ossification was confirmed using sagittal sections stained with modified Goldner's trichrome (Fig. 2z, aa). Likewise, histomorphometric assessment of bone area (B.Ar) confirmed that proximal neurectomy completely abrogated heterotopic bone at this time (Fig. 2a, b, 96% reduction among denervated animals).

Given the prominent reduction of HO formation with surgical denervation, we further examined potential changes in motor function among sham- or neurectomy-treated animals. Overall changes in movement among denervated animals were not conspicuous, and, in fact, video-tracking analysis found a slight increase in overall activity among denervated animals (Supplementary Fig. S2). Thus, peripheral innervation is necessary for the aberrant cell differentiation program that occurs in traumatic HO without significant changes in overall animal activity.

**Soft tissue trauma induces perivascular NGF expression.** Having demonstrated the necessary role of nerve fibers in traumatic HO, we next set out to elucidate the signals that stimulated this nerve ingrowth after injury. NGF is the primary neurotrophin that stimulates skeletal sensory nerve ingrowth[23,24]. To provide detailed dynamic expression analysis, scSeq at the site of HO formation was performed and analyzed prior to injury (day 0) and at the early timepoints of 3, 7, and 21 days after injury in relation to uninjured control (Fig. 3 and Supplementary Fig. S3). Nine cell clusters were defined, which broadly included three mesodermal populations (pericyte/vascular SMC, mesenchymal,

and skeletal muscle precursor), two endothelial populations (endothelial and endothelial lymphatic), and four inflammatory cell populations (macrophage, B cell, T cell, and neutrophil) (Fig. 3a, b and Supplementary Fig. S3). Specific interrogation of *Ngf* expression was performed across cell clusters and across timepoints. A pericyte/vascular SMC population typified by *Acta2* (*actin alpha 2, smooth muscle*) and *Myh11* (*myosin heavy chain 11*) expression represented the primary source of *Ngf* at baseline and after injury (Fig. 3c, d, mean normalized *Ngf* expression of 0.4; *Ngf* detected in 39% of cells). To a much lesser extent, a mesenchymal population represented sources of *Ngf* transcripts, also characterized by the gene markers *Pdgfra* (*platelet-derived growth factor receptor alpha*) and *Prrx1* (*paired related homeobox 1*) (mean normalized *Ngf* expression of 0.04; *Ngf* expressed in 7.4% of cells). Other populations demonstrated rare to an absent expression of *Ngf*, including *Pecam1* (*platelet and endothelial cell adhesion molecule 1*)$^+$ endothelial cells, and individual inflammatory cell populations (cells not belonging to the pericyte/SMC or mesenchymal clusters showed mean normalized *Ngf* expression of 0.009, detected in 1.2% of cells). Subclustering among the highest expressing *Ngf*-expressing pericyte/SMC population confirmed that two subclusters were present (Fig. 3e), which included *Pdgfrb* (*platelet-derived growth factor receptor beta*)$^{high}$ *Prrx1*$^{high}$ *Abcc9* (*ATP-binding cassette, sub-family C, member 9*)$^+$ pericytes and *Acta2*$^{high}$ *Pln* (*phospholamban*)$^+$ *Myh11*$^{high}$ vascular SMCs (Fig. 3f). Of these two subclusters, vascular SMCs demonstrated the highest *Ngf* expression at baseline (for SMCs, mean normalized *Ngf* expression of 0.52; *Ngf* detected in 41% of the cells; for pericytes, mean normalized *Ngf* expression of 0.28; *Ngf* detected in 23% of the cells), and both pericytes and vascular SMCs demonstrated induction of *Ngf* expression after injury (Fig. 3g). Known receptors for NGF were next assessed in the same dataset (Supplementary Fig. S4). The high-affinity receptor neurotrophic receptor tyrosine kinase 1 (*Ntrk1*) encoding TrkA revealed essential absent expression across all sampled cells (Supplementary Fig. S4a, b, mean normalized expression 0.2% of cells). The low-affinity receptor *Ngfr* was primarily expressed in the pericyte/vascular SMC cluster (Supplementary Fig. S4c, d, mean normalized expression 0.3, expressed in 30% of cells), and lowly expressed in all other cell clusters (mean normalized expression 0.017, expressed in 1.6% of cells). Thus, in this model that couples nerve ingrowth and aberrant mesenchymal cell differentiation, perivascular cells demonstrate the highest *Ngf* expression.

To confirm this enrichment in *Ngf* transcripts among distinct cell populations during HO genesis, we next analyzed a

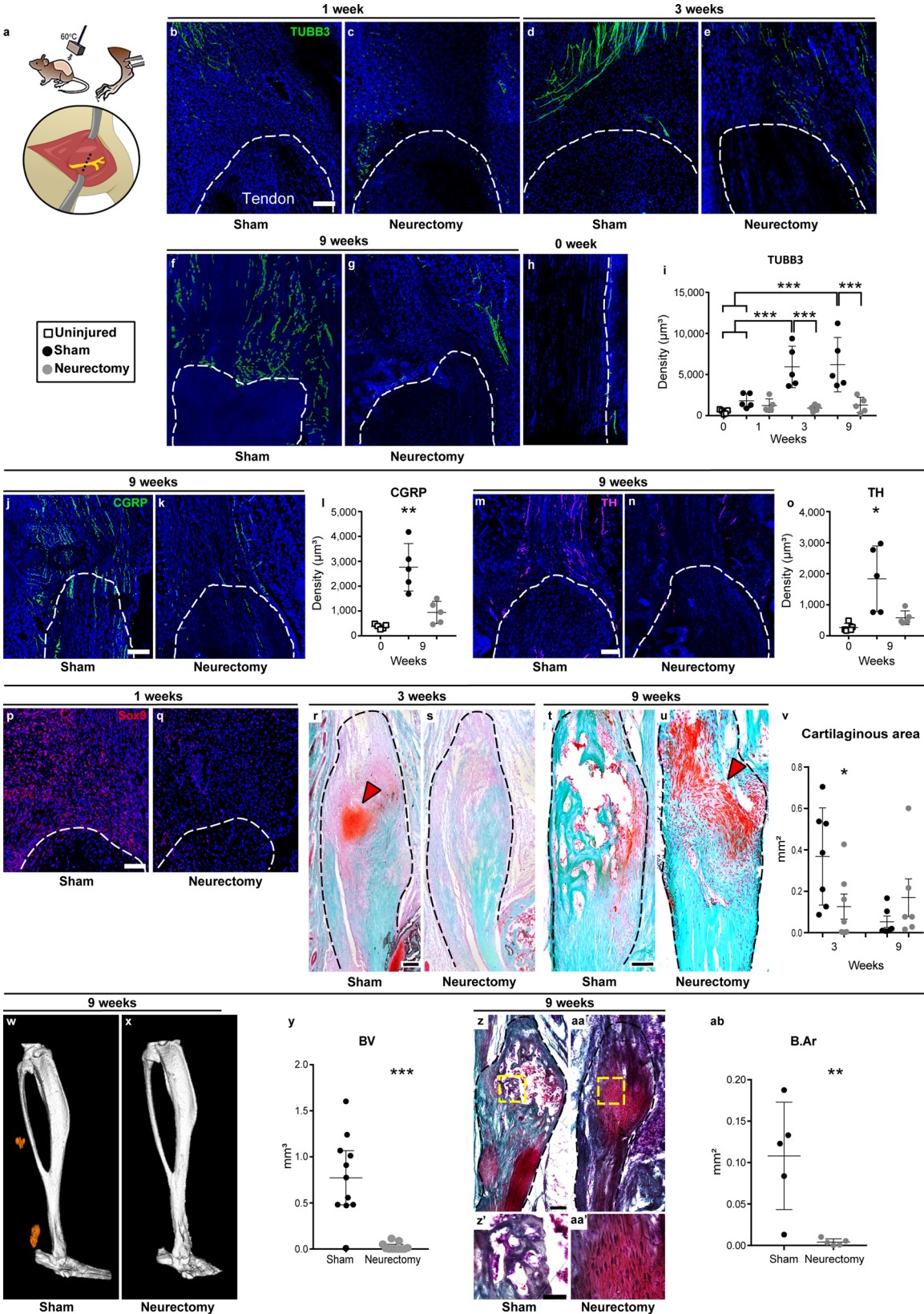

NGF-eGFP reporter animal[37]. This model expresses an enhanced green fluorescent protein (eGFP) driven by the mouse NGF promoter that regulates the transcription for the entire NGF gene, including pro-NGF, mature NGF, and a variety of splice variants from the pre-pro-NGF gene. Here, sagittal cross sections of the transgenic reporter hindlimb were evaluated, focusing on the

distal tenotomy site in comparison to uninjured limbs (Fig. 4a). Within the uninjured Achilles tendon, essentially no NGF reporter activity was observed within the tendon body, and inconspicuous reporter activity was present within the surrounding soft tissue (Fig. 4b). One week after tenotomy and during the fibroproliferative phase, conspicuous NGF reporter activity was

**Fig. 2 Surgical denervation inhibits heterotopic ossification (HO). a** Schematic representation of HO induction, including Achilles tenotomy with dorsal burn (burn/tenotomy) with or without proximal sciatic neurectomy. **b–h** Pan-neuronal Beta III Tubulin (TUBB3) immunohistochemical stains within representative sagittal sections of the distal tenotomy site. Images are shown at 1, 3, and 9 weeks post injury, with concomitant sham or neurectomy surgery, in relation to uninjured control. The dashed white lines indicate the margins of the Achilles tendon. **i** TUBB3$^+$ nerve density at the HO site, among sham or neurectomy groups (***$p < 0.001$). $N = 5$ animals per group. **j, k** Immunohistochemical staining for the peptidergic fiber marker calcitonin gene-related peptide (CGRP) at 9 weeks post injury, neurectomy in relation to sham control. **l** Quantification of CGRP$^+$ nerve density (**$p = 0.0013$). $N = 5$ animals per group. **m, n** Immunohistochemical staining for the sympathetic fiber marker tyrosine hydroxylase (TH) at 9 weeks post injury, neurectomy in relation to sham control. **o** Quantification of TH$^+$ nerve density (*$p = 0.024$). $N = 5$ animals per group. **p, q** SOX9 immunofluorescent stainings at 1 week, among sham or neurectomy groups. Dashed white lines indicate the margins of the Achilles tendon. **r–u** Safranin O/Fast green staining among sagittal cross sections of the distal tenotomy site at 3 and 9 weeks post injury. Dashed black lines indicate the margins of the Achilles tendon. Cartilage appears orange (red arrowhead). **v** Quantification of cartilaginous area within the HO site (*$p = 0.027$). $N = 6$ animals per group, except for 3 weeks Sham group ($N = 7$ animals). **w, x** μCT reconstruction of bone formation within the injured tendon among sham or sciatic neurectomy groups at 9 weeks post injury. Heterotopic bone appears orange, while native bone appears white. **y** μCT quantification of heterotopic bone volume (BV), 9 weeks post injury (***$p < 0.001$). $N = 11$ animals per group. **z, aa** Modified Goldner's trichrome staining among sagittal sections of sham or neurectomy groups, 9 weeks post injury. Dashed yellow boxes represent the high magnification images shown below. **ab** Histomorphometric quantification of bone area (B.Ar) within the HO site (**$p = 0.0071$). $N = 5$ animals per group. Scale bars: 100 μm. For all graphs, each dot represents a single animal, with mean ± 1 SD indicated. Statistical analysis was performed using a two-way Student's $t$ test, with the exception of (**i**), in which a two-sided one-way ANOVA with post hoc Tukey's test was employed. Source data are provided as a Source Data file.

present surrounding the distal tenotomy site (Fig. 4c). During HO progression and at the 3-week timepoint, NGF reporter activity remained elevated in the reactive zone around the tenotomy site (Fig. 4d). Within the osseous phase (9 weeks), NGF reporter activity had waned, but was still elevated above baseline (Fig. 4e). Protein expression of NGF was next evaluated at the injury site by a combination of immunohistochemistry, western blot, and enzyme-linked immunosorbent assay (ELISA) (Supplementary Fig. S5). NGF immunohistochemistry was performed on NGF-eGFP reporter sections, demonstrating a high degree of overlap at 1 week post injury (Supplementary Fig. S5a–c). Western blotting for NGF and pro-NGF confirmed an increase at 7 day post injury (Supplementary Fig. S5d–f). Quantitative analysis of pro-NGF by ELISA confirmed these findings (Supplementary Fig. S5g). Next, the expression of pTrkA was analyzed within the ipsilateral lumbar dorsal root ganglia (DRGs) corresponding to the injured area of the Achilles tendon (Supplementary Fig. S6). Here, a 7.52-fold increase in pTrkA immunoreactivity was observed 48 h post injury.

Immunohistochemical staining of NGF-eGFP reporter sections with cell type-specific antibodies was next performed (Fig. 4f–m). Within sections at 1 week, tenomodulin (TNMD)$^+$ tenocytes were essentially reporter negative (Fig. 4f). Consistent with scSeq findings, α-smooth muscle actin (αSMA)$^+$ perivascular cells showed the highest intensity of NGF reporter activity (Fig. 4g). To a lesser extent, fibroblastic spindle cells not connected to the vasculature demonstrated less strong NGF reporter activity, some of which expressed αSMA (Fig. 4h) and some of which expressed PDGFRα (Fig. 4i). Within this reactive zone, occasional F4/80$^+$ macrophages also demonstrated NGF reporter activity (Fig. 4j). During the progression of HO, the cartilage anlagen was next examined (Fig. 4k, l). At the 3-week timepoint, Aggrecan (ACAN)$^+$, Type X Collagen (Col X)$^+$ chondrocytes showed some weak and nonuniform NGF reporter activity (Fig. 4k, l). At the 9-week timepoint, mature bone was observed, and osteocalcin (OCN)$^+$ osteoblasts and osteocytes showed some reporter activity (Fig. 4m). Thus, using dynamic single-cell transcriptomics as well as specific NGF reporter animals, an acute increase in NGF expression accompanies post-traumatic HO. Moreover, this soft tissue injury results in an increase in local NGF protein expression, as well as increased phosphorylation of TrkA within the DRG whose axons supply this anatomic region.

**Ngf deletion inhibits HO.** We next set out to characterize the requirement of NGF expression for heterotopic bone formation.

For this purpose, *Ngf*$^{fl/fl}$ animals were treated within the injury site with adenovirus-encoding Cre recombinase (Ad-Cre), while *Ngf*$^{fl/fl}$ littermates received Ad-GFP as a vector control (Fig. 5a). Validation of recombination was performed using membranous TdTomato/membranous GFP (mT/mG) reporter animals, in which Ad-Cre demonstrated significant recombination within the tenotomy site at 3 days post operation, appearing green (Fig. 5b, c). In addition, some systemic distribution of adenovirus was confirmed, including most prominently in the lung and liver (Supplementary Fig. S7). Animals with or without *Ngf* deletion were subjected to HO induction, with assessments performed at 9 weeks post injury. The robust axonal invasion was observed under control conditions with Ad-GFP treatment (Fig. 5d, e), which was severely diminished among Ad-Cre-treated *Ngf*$^{fl/fl}$ tenotomy sites (Fig. 5f). The frequency of TUBB3$^+$ axons within the HO site was quantified, demonstrating a mean 757% increase in nerve fiber frequency among Ad-GFP in comparison to uninjured control (Fig. 5g). In contrast, Ad-Cre-treated *Ngf*$^{fl/fl}$ tenotomy sites demonstrated no significant increase in TUBB3$^+$ nerve fiber frequency over baseline (mean 88% reduction in comparison to Ad-GFP control). Findings were confirmed across both CGRP$^+$ sensory fibers and TH$^+$ sympathetic fibers (Fig. 5h–m). CGRP$^+$ nerve ingrowth showed a conspicuous reduction among Ad-Cre-treated *Ngf*$^{fl/fl}$ injury sites (Fig. 5h–j, 73% reduction in comparison to Ad-GFP control). Likewise, TH$^+$ sympathetic fiber ingrowth was blunted among Ad-Cre-treated *Ngf*$^{fl/fl}$ tenotomy sites (Fig. 5k–m, 83% reduction in comparison to Ad-GFP-treated control). As observed in other contexts[24], reduced nerve density may alter vascular ingrowth. To assess injury site vascularity, CD31 immunohistochemical staining was next performed. Ad-Cre-treated *Ngf*$^{fl/fl}$ tenotomy sites demonstrated a marked reduction in CD31$^+$ vascular channels (Supplementary Fig. S8, 81% reduction in comparison to Ad-GFP-treated control). The phenotypic consequence of *Ngf* deletion on bone formation was next assessed. μCT reconstructions demonstrated a significant reduction in ossification among Ad-Cre-treated tenotomy sites in comparison to Ad-GFP-treated controls (Fig. 5n, o), which was confirmed by a significant reduction in BV by μCT analysis (Fig. 5p, mean 60% reduction among Ad-Cre-treated HO sites). A significant reduction in HO was confirmed using sagittal sections stained with modified Goldner's trichrome (Fig. 5q–s). Histomorphometric assessment of B. Ar confirmed that Ad-Cre-treated *Ngf*$^{fl/fl}$ injury sites had significantly less heterotopic bone (Fig. 5s, 63% reduction among Ad-Cre-treated injury sites). Lastly, a significant reduction in

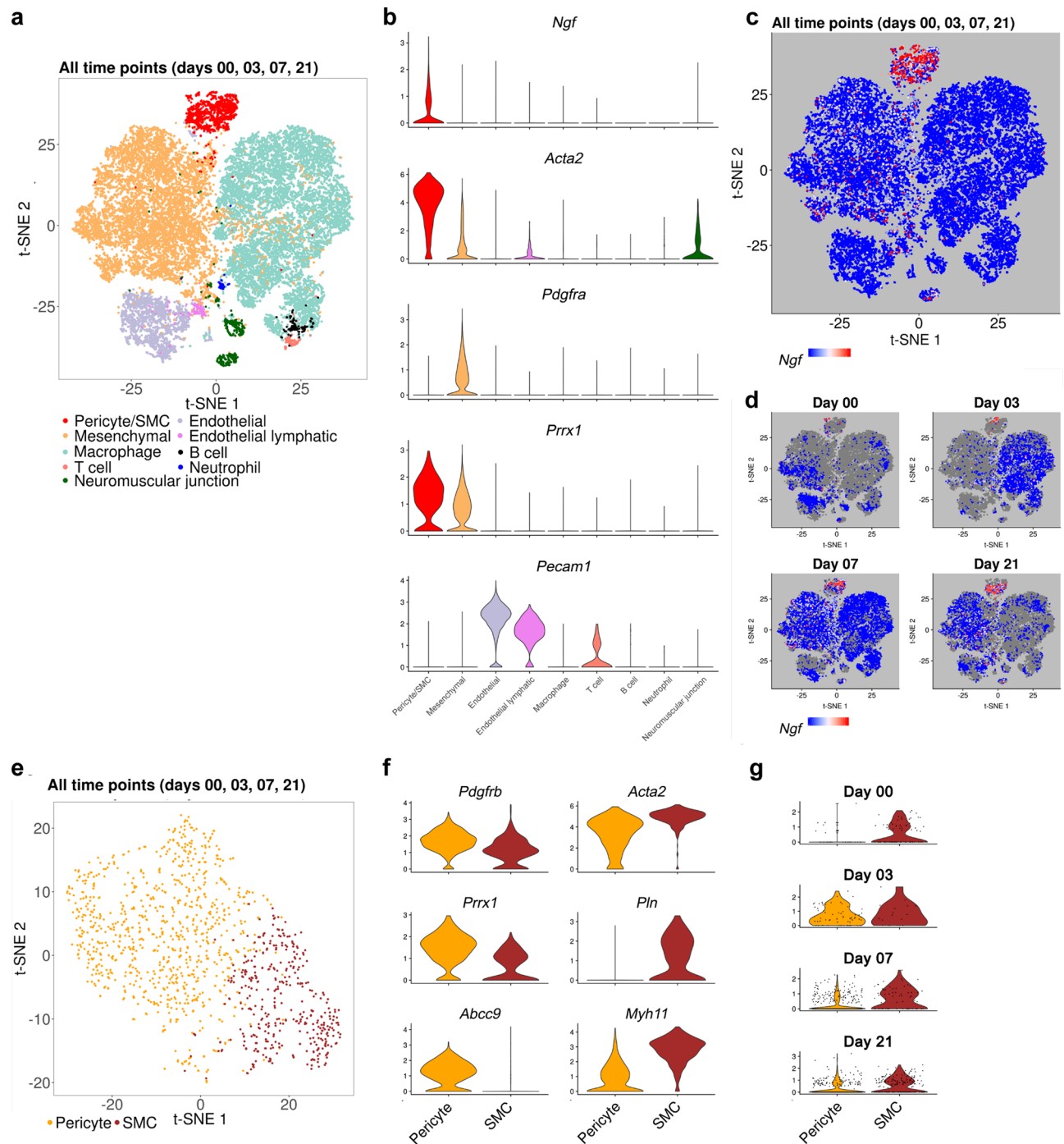

**Fig. 3 The primary sources for *nerve growth factor (Ngf)* during early HO evolution are pericytes and vascular smooth muscle cells (SMCs).** C57BL/6J mice were subjected to HO induction, and the injury site was examined by single-cell RNA-sequencing (scRNA-seq) up to 21 days thereafter. **a, b** t-Distributed stochastic neighbor embedding (t-SNE) plots of pooled cells from four time points after HO induction (including days 0, 3, 7, and 21), along with violin plots to demonstrate characteristic gene markers among nine cell clusters. See Supplementary Fig. S3 for additional gene markers that typify each cluster. **c, d** *Ngf* expression within t-SNE plots either **c** among pooled across all timepoints or **d** at each time point individually. **e** Subclustering among the pericyte/ SMC cell population by t-SNE plot reveals two main cell populations (cells pooled across all timepoints). **f** Genes of interest define the two sub-population as pericytes (yellow) and vascular SMCs (brown). **g** Violin plots demonstrating *Ngf* expression across time in pericyte and vascular SMC subclusters. Both pericytes and SMCs demonstrate *Ngf* expression, with vascular SMCs providing the most basal *Ngf* in uninjured tissue (day 0). Cells isolated from $N = 4$ animals for days 0, 7, and 21; $N = 3$ animals for day 3.

OCN immunoreactivity was observed among Ad-Cre-treated tenotomy sites (Fig. 5t,u). Thus, *Ngf* deletion impairs axonal ingrowth into the injured site, with a concomitant reduction in the pathologic process of trauma-induced heterotopic bone formation, which phenocopies proximal nerve transection.

**Inhibition of TrkA abrogates HO**. Having identified the central role of NGF, we next set to validate that NGF exerts its effects on nerve ingrowth and secondarily stem cell fate decisions via its high-affinity receptor TrkA. First, we used a previously validated chemical–genetic approach, in which TrkA signaling is acutely

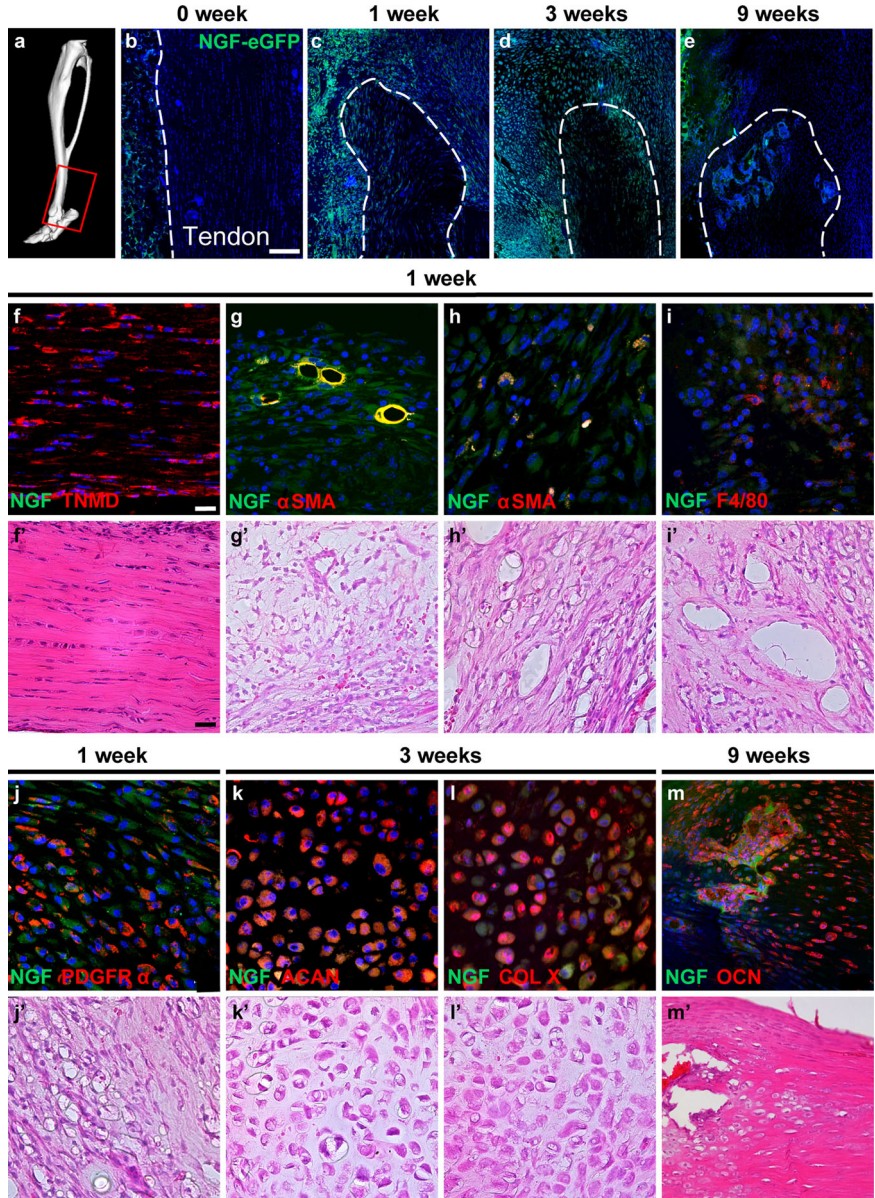

**Fig. 4 NGF-eGFP reporter activity during heterotopic ossification (HO).** NGF-eGFP reporter animals were subjected to HO induction and analyzed at 1, 3, and 9 weeks post injury. **a** μCT reconstruction demonstrating the location of histology at the distal tenotomy site. The red box represents the region of interest. **b–e** Representative tile scans of NGF-eGFP reporter activity within the distal tenotomy site at 1, 3, and 9 weeks post injury. Reporter activity appears green, while nuclear counterstain appears blue. Uninjured control is shown for comparison. Dashed white lines indicate margins of the Achilles tendon. White scale bars: 100 μm. **f–m** Cell sources of NGF, using immunohistochemical analysis of NGF-eGFP reporter sections. For all images, immunohistochemical staining appears red or yellow, while NGF reporter activity appears green. Nuclear counterstain appears blue. Corresponding H&E images are shown below (**f'–m'**). **f–j** Analyses at 1 week post injury, including **f** tenomodulin (TNMD), **g, h** smooth muscle actin (SMA) among perivascular cells (**g**) and non-vascular fibroblastic cells (**h**), **i** platelet-derived growth factor receptor-α (PDGFRα), and **j** F4/80. **k, l** Analyses at 3 weeks post injury, including **k** Aggrecan (ACAN) and (**l**) Type X Collagen (Col X). **m** Analysis at 9 weeks post injury, including osteocalcin (OCN). White and black scale bars: 50 μm. $N = 3$ animals per timepoint.

disrupted over a defined period of time. TrkA^F592A mice are homozygous for knock-in alleles that encode a phenylalanine-to-alanine substitution in the protein kinase subdomain V, rendering its catalytic activity sensitive to specific inhibition by the membrane-permeable, small-molecule 1NMPP1[38]. Previously, we have validated the temporal kinetics of 1NMPP1 inhibition in TrkA^F592A animals and verified that 1NMPP1 has no discernible direct effects on osteoblastic cells[23]. Moreover, we confirmed that 1NMPP1-treated TrkA^F592A mice showed no discernible phenotype of the uninjured Achilles tendon (Supplementary Fig. S9).

TrkA^F592A mice or TrkA^WT littermates were now subjected to HO induction by burn/tenotomy, and all animals provided 1NMPP1 (Fig. 6a). von Frey testing confirmed a sensory deficit within TrkA^F592A mice (Supplementary Fig. S1b, 38% delay in paw withdrawal in comparison to 1NMPP1-treated TrkA^WT littermates). As expected, TrkA^F592A mice demonstrated a reduction in nerve fiber frequency (Fig. 6b–g and Supplementary Fig. S10). This was first observed at baseline within the uninjured peritenon of TrkA^F592A animals (Fig. 6b, e and Supplementary Fig. S10). As expected, TrkA^WT animals demonstrated a robust invasion of TUBB3+ nerve fibers within

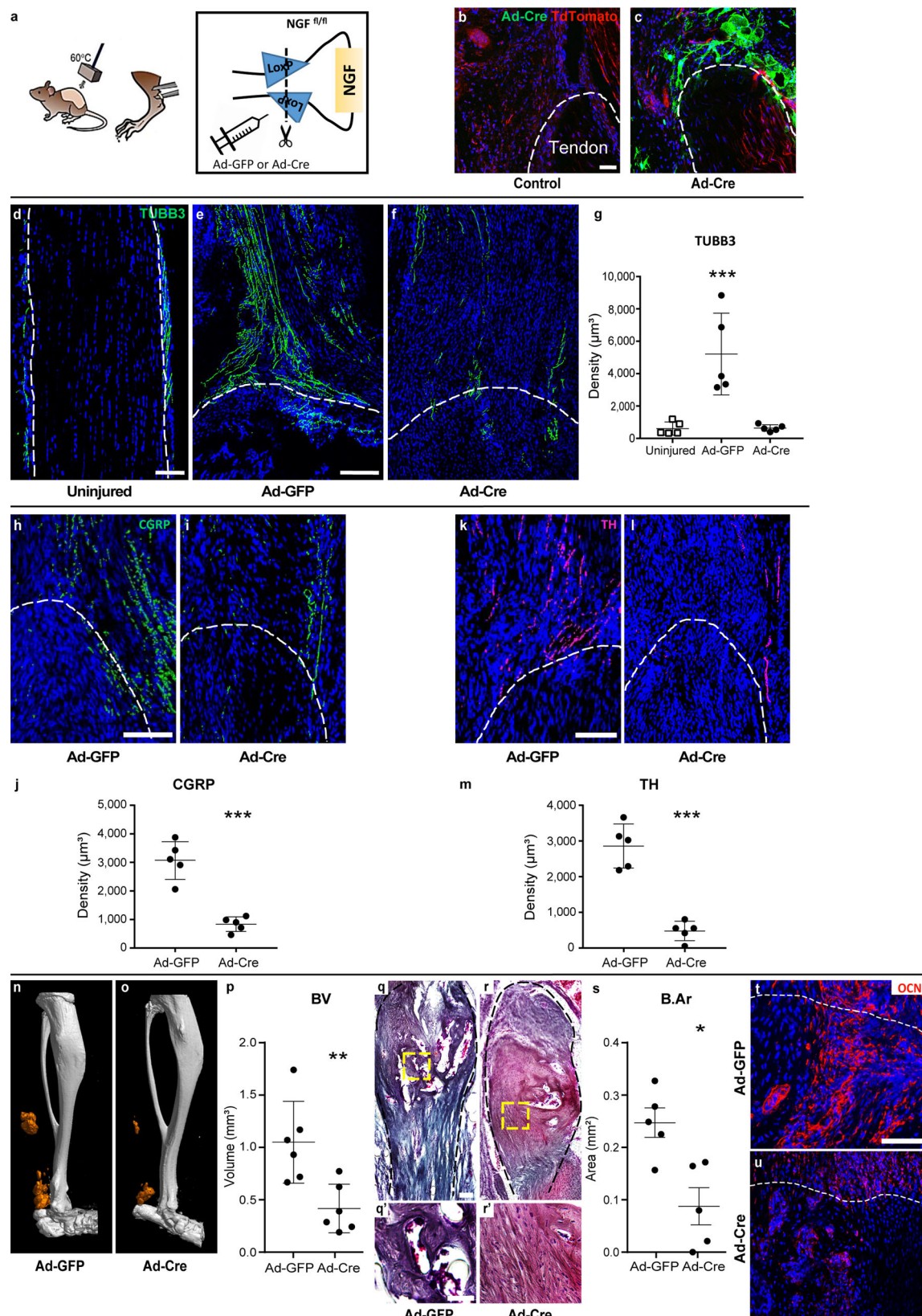

the distal tenotomy site (Fig. 6c, d), which was significantly blunted among TrkA[F592A] littermates (Fig. 6f, g). Quantitative histologic analysis was next performed using TUBB3-immunostained sections (Fig. 6h). As in prior experiments, an increase in TUBB3[+] nerve fiber density was observed among TrkA[WT] injury sites (up to 442% increase during 9 weeks),

while no significant increase in nerve fiber frequency was observed within TrkA[F592A] animals (Fig. 6h).

The effects of inhibition of TrkA catalytic activity on cartilage and bone formation in HO progression was next assessed. Sagittal sections of the injured tendon were stained with Safranin O to detect cartilage within the developing HO site (Fig. 6i–m). At

**Fig. 5 Nerve growth factor (Ngf) deletion abrogates heterotopic ossification (HO). a** Schematic representation of thhe experiment. Mice with Ngf floxed alleles (Ngf [fl/fl]) underwent local injection with Ad-Cre or Ad-GFP control, followed by HO induction. **b, c** Validation of Cre-recombination at the HO site among Ad-Cre-injected animals. Ad-Cre injection was performed within mT/mG reporter animals, followed by burn/tenotomy. Cells with Cre-mediated recombination appear green, 3 days post injury. Dashed lines indicate margins of the tendon. **d–f** TUBB3 (Beta III Tubulin) immunofluorescent staining among Ad-GFP or Ad-Cre injected Ngf [fl/fl] distal tenotomy sites, at 9 weeks post injury. **g** Quantification of TUBB3[+] nerve density among Ad-GFP or Ad-Cre injected Ngf[fl/fl] HO sites, at 9 weeks post injury in relation to uninjured control (***$p < 0.001$). $N = 5$ animals per group. **h, i** Immunohistochemical staining for calcitonin gene-related peptide (CGRP) at 9 weeks post injury among Ad-GFP- versus Ad-Cre-treated tenotomy sites. **j** Quantification of CGRP[+] nerve density (***$p < 0.001$). $N = 5$ animals per group. **k, l** Immunohistochemical staining for tyrosine hydroxylase (TH) at 9 weeks post injury among Ad-GFP- versus Ad-Cre-treated tenotomy sites. **m** Quantification of TH[+] nerve density (***$p < 0.001$). $N = 5$ animals per group. **n, o** μCT reconstruction of bone formation within the injured tendon among Ad-GFP- or Ad-Cre-treated groups at 9 weeks post injury. Heterotopic bone appears orange, while native bone appears white. **p** μCT quantification of heterotopic bone volume (BV), 9 weeks post injury (**$p = 0.0066$). $N = 6$ animals per group. **q, r** Modified Goldner's trichrome staining among sagittal sections of Ad-GFP- or Ad-Cre-treated Ngf[fl/fl] injury sites, 9 weeks post injury. Dashed yellow boxes represent the high magnification images shown below. **s** Histomorphometric quantification of bone area (B.Ar) within the HO site (*$p = 0.021$). $N = 5$ animals per group. **t, u** Osteocalcin (OCN) immunohistochemical staining, appearing red, at 9 weeks post injury. For all graphs, each dot represents a single animal, with a mean ± 1 SD indicated. Statistical analysis was performed using a two-way Student's t test. Scale bars: 100 μm. Source data are provided as a Source Data file.

3 weeks post injury, an accumulation of stained cartilage glycosaminoglycans was observed among TrkA[WT] injury sites, which was not found among TrkA[F592A] littermates (Fig. 6i, j). At 9 weeks, frank bone formation was observed within TrkA[WT] tissues, while cartilage was now observed among TrkA[F592A] hindlimbs (Fig. 6k, l). These findings of delayed endochondral ossification were again confirmed using histomorphometric analysis of serial sections to delineate C.Ar (Fig. 6m). Under TrkA[WT] conditions, cartilaginous tissue areas were prominent at 3 weeks and barely detected at 9 weeks as endochondral ossification progressed. In contrast and in TrkA[F592A] specimens, cartilaginous tissue was much reduced at 3 weeks (Fig. 6m, mean 83% reduction in C.Ar in TrkA[F592A] versus TrkA[WT] limbs at the 3-week timepoint, $p = 0.06$). Residual cartilage was confirmed at 9 weeks among TrkA[F592A] animals (mean 243% increase in C.Ar in comparison to TrkA[WT] animals). This delay in endochondral ossification among TrkA[F592A] injury sites was confirmed using μCT, demonstrating a reduction in heterotopic bone visualized by μCT reconstructions (Fig. 6n, o). The quantitative μCT assessment of BV confirmed that TrkA[F592A] animals demonstrated a significant reduction in heterotopic bone (Fig. 6p, 74% reduction within TrkA[F592A] injury sites). Reduction in extraskeletal ossification was next confirmed using sagittal sections stained with modified Goldner's trichrome (Fig. 6q, r). Histomorphometric assessment of B.Ar verified that TrkA[F592A] mice had notably less heterotopic bone compared to TrkA[WT] mice (Fig. 6s, mean 52% reduction among TrkA[F592A] injury sites).

To further define the translational potential of TrkA inhibition for the prevention of HO, we utilized a small-molecule antagonist in clinical development, AR786, and applied this to C57BL/6J wild-type mice (Fig. 7). AR786 is a selective intracellular TrkA kinase activity inhibitor, which has an oral bioavailability[39,40] (Fig. 7a). AR786-treated animals revealed a decline in TUBB3[+] nerve fiber density as compared to vehicle control-treated mice (Fig. 7b–e). AR786-induced reduction of nerve fiber density was observed both within the uninjured peritenon (Fig. 7b, c) and within the injury site (Figs. 7d, e, 3 weeks post surgery). Quantitative analysis indicated an increase in TUBB3[+] nerve fiber density among vehicle-treated mice injury sites (up to 494% increase), whereas no significant increase in nerve fiber frequency was observed within AR786-treated mice (Fig. 7f, 85% reduction in comparison to vehicle control at corresponding timepoint). Safranin O staining within sagittal sections of the injury site demonstrated an accumulation of cartilage glycosaminoglycans in control conditions, which was essentially not found in AR786-treated animals (Fig. 7g, h). Histomorphometric analysis indicated that AR786-treated mice showed 69% reduction in C.

Ar in comparison to control (Fig. 7i). A similar pattern was detected in immunostaining for cartilage antigens, including SOX9 and COL2 (Fig. 7j–m). Increased immunostaining at the tenotomy site was observed among control-treated animals, but not in AR786-treated animals. Thus, TrkA inhibition via two independent methods impairs axonal ingrowth into the injured soft tissues, with a parallel reduction in trauma-induced heterotopic cartilage and bone.

**Denervation shifts TGFβ to FGF signaling activation in pre-chondrogenic cells.** To better understand the downstream molecular mechanisms whereby impaired axonal ingrowth prevents aberrant osteochondral differentiation, scSeq was again performed, now in the context of sham surgery or ipsi-lateral sciatic neurectomy (Fig. 8a). Seven days after an HO inducing injury with or without proximal neurectomy, tissue was isolated for scSeq and dimensional reduction resulted in 10 unique clusters (Fig. 8b) identified by expression of unique cell markers (Fig. 8c). Overall similarities were found to our prior dataset, including several mesodermal clusters including Col1a1-expressing mesenchymal clusters (clusters 2 and 5, also found to highly express Pdgfra), as well as an Acta2-expressing pericyte/SMC cluster. As in our prior dataset, expression of Ngf was again principally confined to the pericyte/SMC cluster and was unchanged by denervation (Supplementary Fig. S13). Both sham and neurectomy samples were represented in all clusters (Supplementary Fig. S12a), and cell frequencies were overall similar between sham and neurectomy conditions (Supplementary Fig. S12b).

Mesenchymal clusters were isolated and reanalyzed, generating three unique subclusters (Fig. 8d). These clusters were identified as Tppp3-expressing tendon sheath-like (TSL), Tnc-expressing tenocyte-like (TL), and a third population enriched with cartilage markers such as Comp and Thbs4, which we designated fibrocartilage-like (FCL). TSL and TL cells shared gene signatures with similar cluster designations recently reported in a mouse patellar tendon repair model[41]. While TSL and TL subclusters were equally represented in both the sham and neurectomy groups, the FCL subcluster was enriched in neurectomy animals (batch size corrected, 72.6% of total) (Fig. 8e). Cell cycle analyses revealed that within the FCL subcluster, neurectomy notably reduced cell proliferation (Fig. 8f).

To better understand putative axon-to-mesenchymal cell para-crine relationships in our model, ligand–receptor mapping was performed using two single-cell datasets. All potential ligands were identified from peptidergic nociceptive fibers, using a previously generated scSeq dataset derived from lumbar dorsal root ganglia[42].

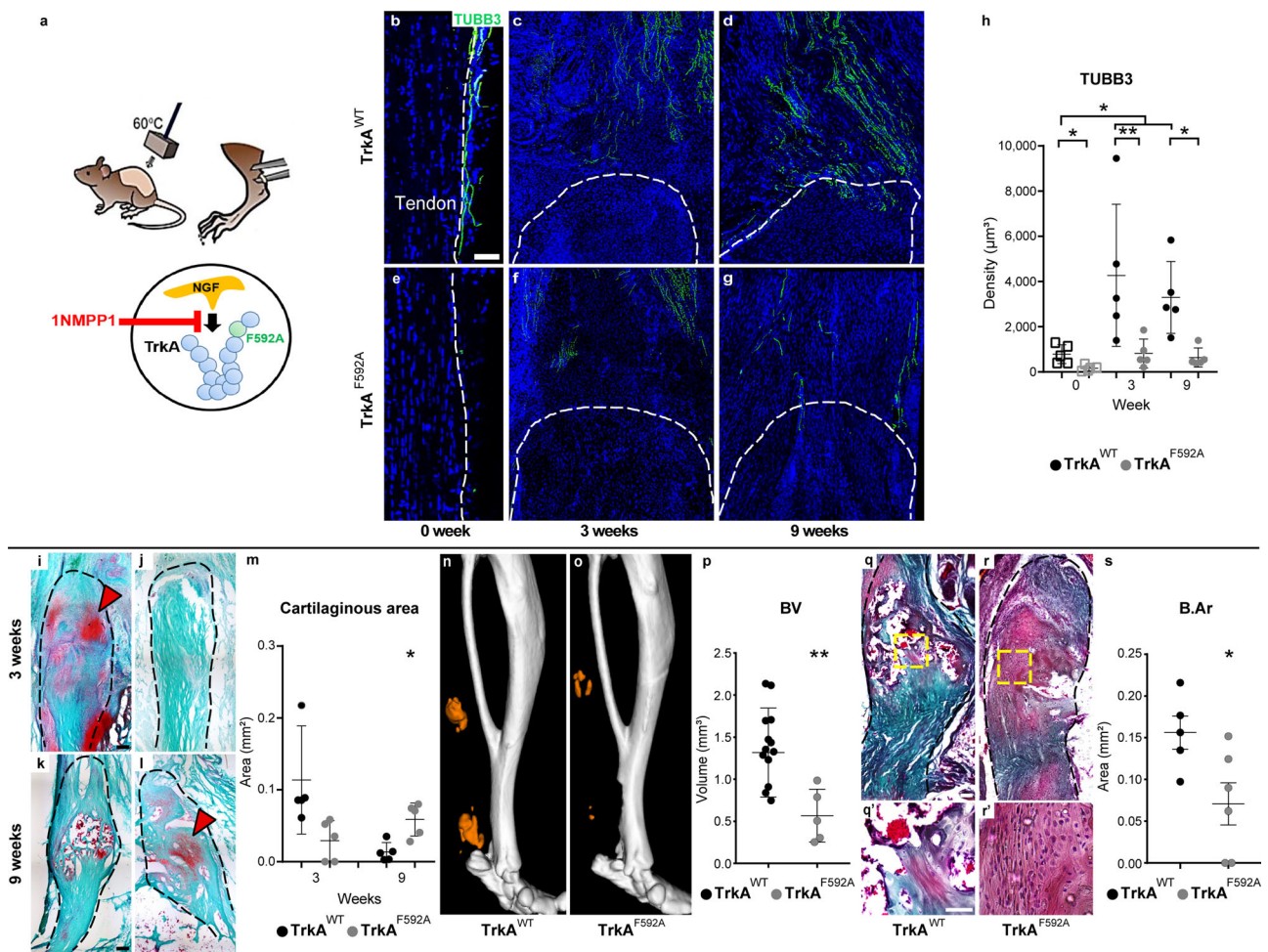

**Fig. 6 TrkA^F592A mice inhibits heterotopic ossification (HO). a** Schematic representation of the experiment. TrkA^F592A mice have a point mutation which renders them susceptible to temporally controllable inhibition of TrkA catalytic activity, via administration of the small-molecule 1NMPP1. Either TrkA^WT or TrkA^F592A animals were subjected to HO induction and both administered 1NMPP1 throughout the study period. **b–g** Beta III Tubulin (TUBB3) immunofluorescent staining of the distal tenotomy site as visualized using sagittal cross sections. Dashed white lines indicate edges of the Achilles tendon. **h** Quantification of TUBB3^+ nerve density among HO sites of TrkA^WT or TrkA^F592A animals, at 3 and 9 weeks post injury in comparison to uninjured control (uninjured contralateral limb denoted as 0 week) (uninjured TrkA^WT versus uninjured TrkA^F592A, *p = 0.049; uninjured TrkA^WT versus 3 weeks TrkA^WT, *p = 0.012; uninjured TrkA^WT versus 9 weeks TrkA^WT, *p = 0.029; 3 weeks TrkA^WT versus 3 weeks TrkA^F592A, **p = 0.008; 9 weeks TrkA^WT versus 9 weeks TrkA^F592A, *p = 0.013). N = 5 animals per group. **i–l** Safranin O/Fast Green staining among the distal tenotomy site of TrkA^WT or TrkA^F592A animals, 3 and 9 weeks post injury. Dashed black lines indicate the margins of the Achilles tendon. Cartilage appears orange (red arrowhead). **m** Quantification of cartilaginous area within the HO site among TrkA^WT or TrkA^F592A animals (*p = 0.035). N = 5 animals per group. **n, o** μCT reconstruction of bone formation within the injured tendon among TrkA^WT or TrkA^F592A animals at 9 weeks post injury. Heterotopic bone appears orange, while native bone appears white. **p** μCT quantification of bone volume (BV) among TrkA^WT or TrkA^F592A animals, 9 weeks post injury (**p = 0.005). N = 13 TrkA^WT animals, N = 5 TrkA^F592A animals. **q, r** Modified Goldner's trichrome staining among TrkA^WT or TrkA^F592A animals, 9 weeks post injury. Dashed yellow boxes represent the high magnification images shown below. **s** Histomorphometric quantification of bone area (B.Ar) within the HO site among TrkA^WT or TrkA^F592A animals, 9 weeks post injury (*p = 0.041). N = 5 TrkA^WT animals and N = 6 TrkA^F592A animals. Scale bars: 100 μm. For all graphs, each dot represents a single animal, with mean ± 1 SD indicated. Statistical analysis performed using a two-sided one-way ANOVA with post hoc Tukey's test (**h**) or two-way Student's t test (**m**, **p**, and **s**). Source data are provided as a Source Data file.

In descending frequency, secreted ligands were cross-referenced to confirm the expression of their cognate receptor(s) within mesenchymal cells of our dataset. This generated a list of 150 ligand–receptor pairs (Supplementary Fig. S11), of which 48% were growth and differentiation factors, 27% were chemokines, and 25% were neuropeptides. Selected pathways that potentially modulate chondrogenic differentiation[30] and that have been shown to play a role in nerve–bone crosstalk are shown in Fig. 8g, and included FGF, TGFβ, PDGF, VEGF, and WNT signaling[43].

Multiple signaling pathways may be important in axon-to-mesenchymal cell paracrine interaction during HO genesis; we next analyzed signaling activation of each pathway in the context

of denervation. For this purpose, we returned to the FCL subcluster, the cell cluster that demonstrated a pre-chondrogenic gene signature after injury, and examined differences in pathway activation among sham-operated versus neurectomy groups. Average expression analyses of pathway gene lists demonstrated upregulation of FGF signaling activation and a simultaneous decrease in TGFβ signaling activation in FCL cluster cells following neurectomy (Fig. 8h). Upregulation of FGF signaling following neurectomy was typified by upregulation of several genes such as *Fgf2*, *Fgf11*, *Frs2*, and *Fgfr2* (Supplementary Table S1). Downregulation of TGFβ signaling-associated genes after neurectomy included, for example, both *Tgfβ2* and *Tgfβ3*, as

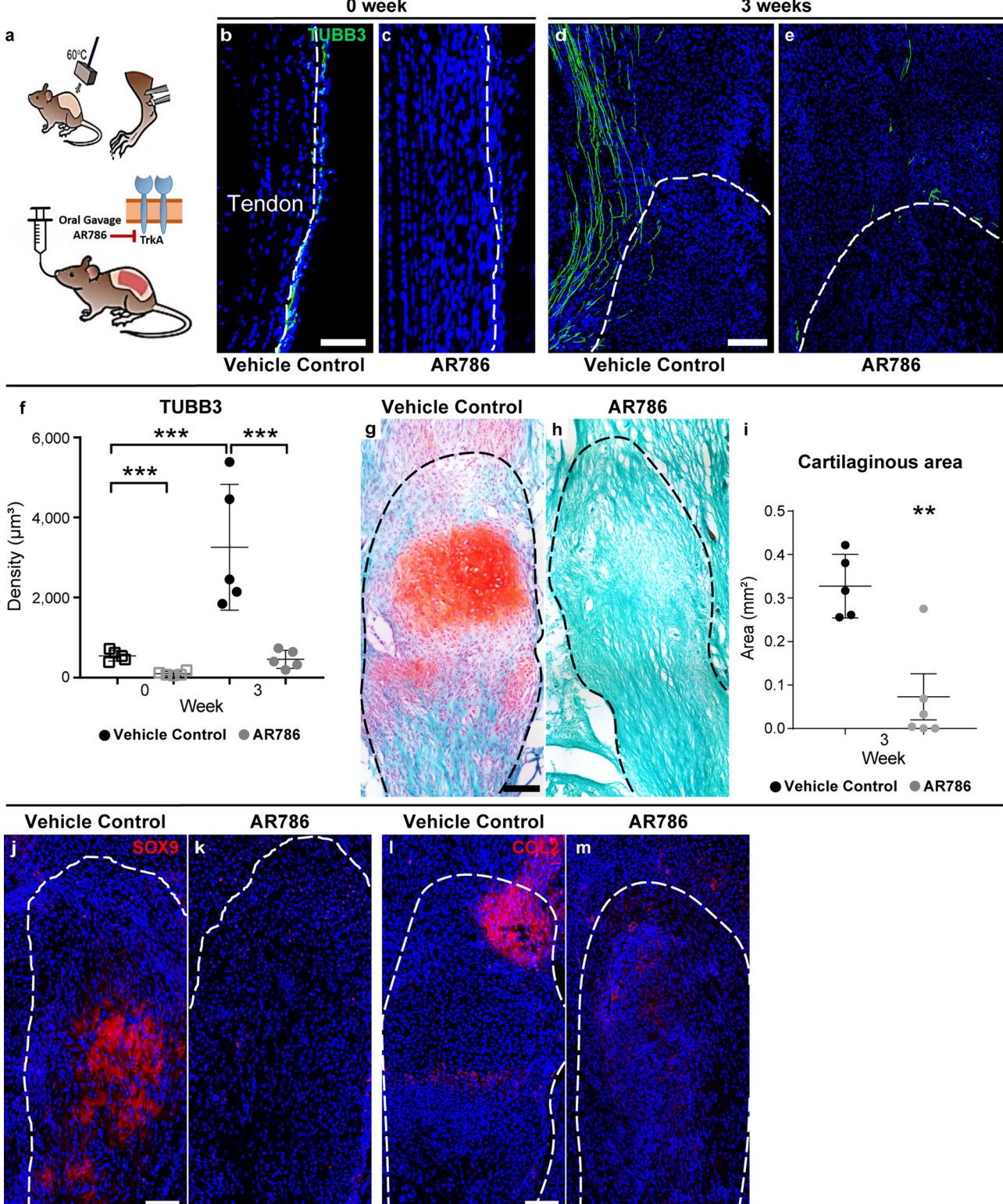

well as *Thbs1* and *Smad3*. Other pathways such as WNT, VEGF, and PDGF showed more modest changes across groups (Fig. 8h). To further investigate a shift from TGFβ to FGF signaling activation, immunohistochemistry for pSmad2, FGF2, and pERK1/2 at the injury site was examined with or without surgical denervation (Fig. 8i–n). Results further supported a shift from TGFβ to FGF signaling activation among denervated injury sites, including a reduction in pSmad2 immunostaining (Fig. 8i, j), and

an increase in FGF2 and pERK1/2 immunostaining (Fig. 8k–n). Thus, loss of neural input is associated with a shift from TGFβ to FGF signaling activation, potentially explaining the loss of chondrogenic differentiation.

Secondary effects of neurectomy on other non-mesenchymal cell populations were next assessed. Neutrophils were noted to slightly segregate between sham and neurectomy conditions (Supplementary Fig. S12c). Isolation and reanalyses of this neutrophil cluster

**Fig. 7 TrkA inhibition using orally bioavailable small-molecule AR786 inhibits heterotopic ossification (HO) in WT mice. a** Schematic representation of the experiment. WT mice were administered AR786 or vehicle control throughout the study period. HO induction was performed (burn/tenotomy), with analysis at 3 weeks. **b–e** TUBB3$^+$ immunofluorescent staining of the distal tenotomy site, as visualized using sagittal cross sections. Dashed white and black lines indicate edges of the Achilles tendon. **f** Quantification of TUBB3$^+$ nerve density among HO sites of vehicle control- or AR786-treated animals, 3 weeks post injury in comparison to the uninjured tendon (uninjured vehicle control versus uninjured AR786, ***$p = 0.0003$; uninjured vehicle control versus 3 weeks vehicle control, ***$p = 0.0001$; 3 weeks vehicle control versus 3 weeks AR786, ***$p = 0.0002$. $N = 5$ animals per group). **g, h** Safranin O/ Fast Green staining among the distal tenotomy site of vehicle control- or AR786-treated animals, at 3 weeks post injury. Cartilage appears orange in Safranin O/Fast Green staining. **i** Quantification of cartilaginous area within the HO site among vehicle control- or AR786-treated animals at 3 weeks post injury (**$p = 0.0038$). $N = 5$ animals in vehicle control, $N = 6$ animals in AR786. **j, k** SRY-Box transcription factor 9 (SOX9) immunostaining within injured tendon among vehicle control- or AR786-treated animals, 3 weeks post injury. **l, m** Collagen 2 (COL2) immunofluorescent staining of the distal tenotomy site of vehicle control- or AR786-treated mice, 3 weeks post injury. Scale bars: 100 μm. For all graphs, each dot represents a single animal, with mean ± 1 SD indicated. Statistical analysis was performed using a two-sided one-way ANOVA with post hoc Tukey's test (**f**) or two-way Student's $t$ test (**i**). Source data are provided as a Source Data file.

showed that DEGs enriched for GO terms associated with increased immune cell migration and chemotaxis in sham neutrophils versus neurectomy (Supplementary Fig. S12d). Macrophages were also subjected to additional analyses (Supplementary Fig. S12e). Although changes were relatively modest, macrophages derived from neurectomy animals showed a shift in gene signature with a reduced expression of M1-related genes and a concomitant increase in M2 genes (Supplementary Fig. S12f). Of note and based on our prior studies[44], a shift from M1- to M2-related genes would not be consistent with a direct causative role in HO reduction. Acute inflammation was next assayed at 1 week post injury using flow cytometry for quantitative assessments of inflammatory cell frequencies (Supplementary Fig. S14). Flow cytometry for inflammatory cell populations confirmed these findings, including no difference in frequency of CD11b$^+$ leukocytes, F4/80$^+$ macrophages, Ly6g$^+$ neutrophils, and minor differences in Ly6c$^{high}$- and Ly6c$^{low}$-expressing monocytes with or without surgical denervation (Supplementary Fig. S14). Thus, minor differences in gene expression of populations of local myeloid cells were observed after denervation, including granulocytes and monocytes, which may play secondary roles in impacting stem cell fate after injury.

**NGF expression and axonal invasion is associated with human HO.** Finally, the relevance of NGF-TrkA signaling and heterotopic bone formation in human patients was assessed. Ossification of the posterior longitudinal ligament (OPLL) is a fairly prevalent degenerative process of the supporting connective tissue of the spine, where ectopic ossifications form within ligamentous tissue. Previous literature explored the effects of cyclical mechanical strain on ligamentous cells harvested from OPLL patients[45]. Re-examination of this microarray dataset (GSE5464) demonstrated significant enrichment of *Ngf* in cultured ligament cells from OPLL patients relative to the control group (Fig. 9a, log fold change 5.48, $p = 0.02$). Associated downstream NGF signaling members (such as *MAPK8*, *MAPK14*, *PTGER4*, and *PENK*) were also increased among OPLL patient cells.

Next, NGF expression was further assayed across human tissue sections of excised HO (Fig. 9b–e). NGF immunoreactivity was present in several mesodermal cell types, including perivascular cells adjacent to bone (Fig. 9b, c). Among samples in the earliest fibroproliferative phases and before the appearance of frank bone, NGF immunoreactive cells were present within cells of fibro-blastic morphology throughout the involved areas (Fig. 9d). In more mature clinical examples of HO with frank bone formation, osteoid seams and bone-associated cells demonstrated robust NGF immunoreactivity (Fig. 9b, e). Concordantly, fine nerve fibers were found at an increased frequency within the soft tissues surrounding HO excision specimens examined by routine hematoxylin and eosin (H&E) staining (Fig. 9f, g). Immunohis-tochemical staining for the neural antigen S100 and

neurofilament SM31 helped to highlight disorganized axonal sprouting at the edges of human HO (Fig. 9h, i). Thus, human HO mirrors key elements of our experimental findings, including NGF expression among mesodermal cell types accompanied by aberrant axonal ingrowth among HO-afflicted tissues.

## Discussion

In this study, we used a combination of surgical and transgenic animal models to demonstrate the central importance of NGF-responsive TrkA-expressing nerves in the response to soft tissue trauma and subsequent aberrant stem cell differentiation to form cartilage and bone. This appears to involve NGF-expressing vascular wall-resident cells that induce neural ingrowth at incipient sites of chondro-osseous differentiation, which in turn positively regulates pathologic endochondral ossification within soft tissues. Other groups have previously examined the potential endoneurial cellular contribution to HO formation[31–33]. Importantly, this endoneurial-derived HO does not agree with prior lineage tracing studies performed by our laboratory and others[46,47]. Instead, our study demonstrates the potential central paracrine relationships between pericytes, TrkA-expressing axons, and mesenchymal progenitor cells in aberrant mesench-ymal stem cell differentiation after trauma.

An important question that remains to be further assessed is the downstream molecular mechanisms by which invading axons positively regulate the evolution of HO. In particular, the secondary messengers released by axons which positively regulate cartilage or bone formation in soft tissues. In amphibian limb regeneration, numerous nerve-derived molecules have been explored including anterior gradient protein, FGFs, BMPs, and glial growth factor[26,48–51]. Using receptor–ligand matching, our experiments suggested that diverse growth and differentiation factors are present at the transcriptional level with DRG-derived sensory axons, and their cognate receptors were present with mesenchymal cells at the injury site. Of these, a shift from TGFβ signaling activation to FGF signaling activation was most apparent in the context of surgical denervation. The importance of TGFβ signaling in experimental HO formation has been described by our group and others[44,52,53], and a loss of mesenchymal TGFβ signaling activation among denervated tissues implicates this pathway as a putative nerve-derived growth factor that drives HO formation. The activation of FGF signaling following neurectomy may have a putative effect on suppressing an osteochondrogenic program after injury. The critical significance of FGF family in normal bone development and HO formation has been documented[54,55]. Conversely, augmented FGF signaling activation in neurectomized cells may inhibit the osteo-chondrogenic differentiation of MSC, which would be consistent with our overall findings[56]. Alternatively, upregulation of FGF signaling following neurectomy may represent a prolonged fibro-proliferative or cartilaginous phase of HO[56].

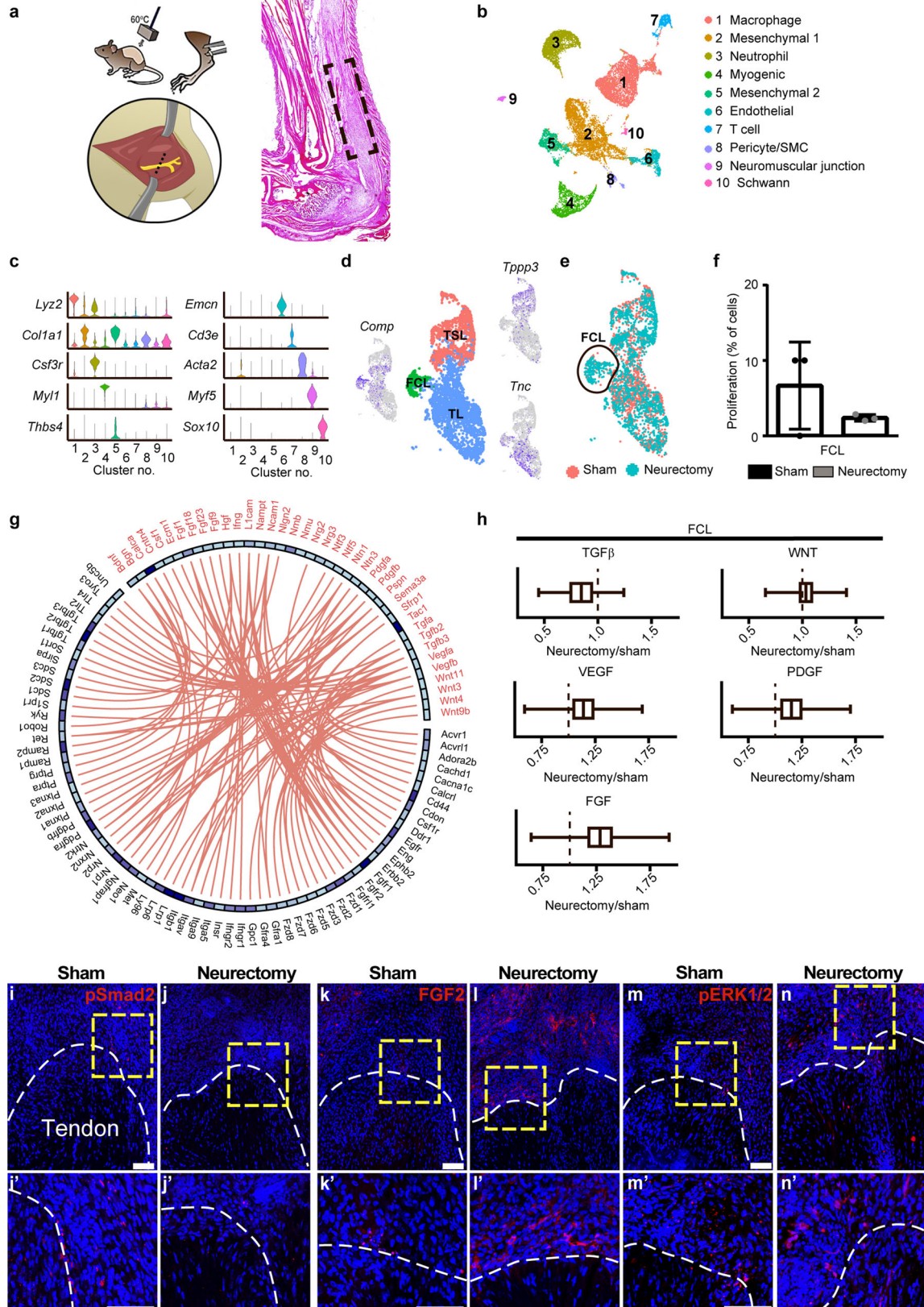

Beyond FGF and TGFβ signaling, several other signaling pathways were considered within the present model. Substance P (SP) expression in multiple animal HO models and human samples has been well documented by immunohistochemistry[33,57,58], and exogenous delivery of SP induces tendon-associated HO[59]. However, our single-cell analysis did not support significant *Tac1* expression

(encoding SP) nor the receptor *Tacr1* in the local HO site. Of note, axons are not sampled using scRNA-seq, and the extent to which peptidergic axon-derived SP modulates HO in the present model needs to be further addressed. Other investigators have observed that Hedgehog signaling plays a well-established role in HO development[60]. In addition, Hedgehog ligands released from sensory

**Fig. 8 Single-cell RNA-sequencing identifies shifts in chondrogenic cell signaling after neurectomy. a** Schematic representation of HO induction with or without proximal sciatic neurectomy (left), followed by microdissection of injury site after 7 days (right). Dashed black box represents the area of tissue dissection for single-cell RNA-sequencing. **b**, **c** UMAP projection and clustering of pooled HO-derived cells from each condition at day 7 post injury, along with violin plots to demonstrate characteristic gene markers among ten cell clusters. **d** Subclustering of mesenchymal cells reveals three cell clusters, shown by UMAP projection (center) with representative genes shown. TSL tendon sheath-like, TL tenocyte-like, and FCL fibrocartilage-like. *Tppp3* tubulin polymerization promoting protein family member 3, *Tnc* tenascin C, and *Comp* cartilage oligomeric matrix protein. **e** UMAP showing the distribution of sham and neurectomy samples across mesenchymal subcluster. **f** Proliferative index among fibrocartilage-like cells within sham or neurectomy conditions. **g** Receptor–ligand mapping for potential paracrine factors released from peptidergic nerve fibers (ligand) and mesenchymal cells (receptor). Ligands derived from among the top 275 ligands ranked by gene expression. Receptors derived from mesenchymal cell cluster from Fig. 3. **h** Signaling pathway activation scores among fibrocartilage cells under neurectomy versus sham conditions, including TGFβ, WNT, VEGF, PDGF, and FGF signaling. **i–n** pSmad2, FGF2, and pERK1/2 immunofluorescent staining of the distal tenotomy site, shown in sagittal cross section. Analysis performed 1 week after injury, with or without concurrent sciatic neurectomy groups. Dashed white line shows the edge of the Achilles tendon and dashed yellow boxes represent the high magnification images shown below. Graphs represent mean values (vertical line), 1st and 3rd quartiles (box), and standard deviation (whiskers) for the combined pathway gene list. Values >1 indicates increased signaling activation in neurectomy groups, while values <1 indicates increased relative expression among the sham group. Cells isolated from $N = 3$ animals per group from a single experimental replicate. See corresponding Supplementary Fig. S12 for additional analysis. Statistical analysis was performed using a two-way Student's *t* test (**f**). Scale bar: 100 µm. Source data are provided as a Source Data file.

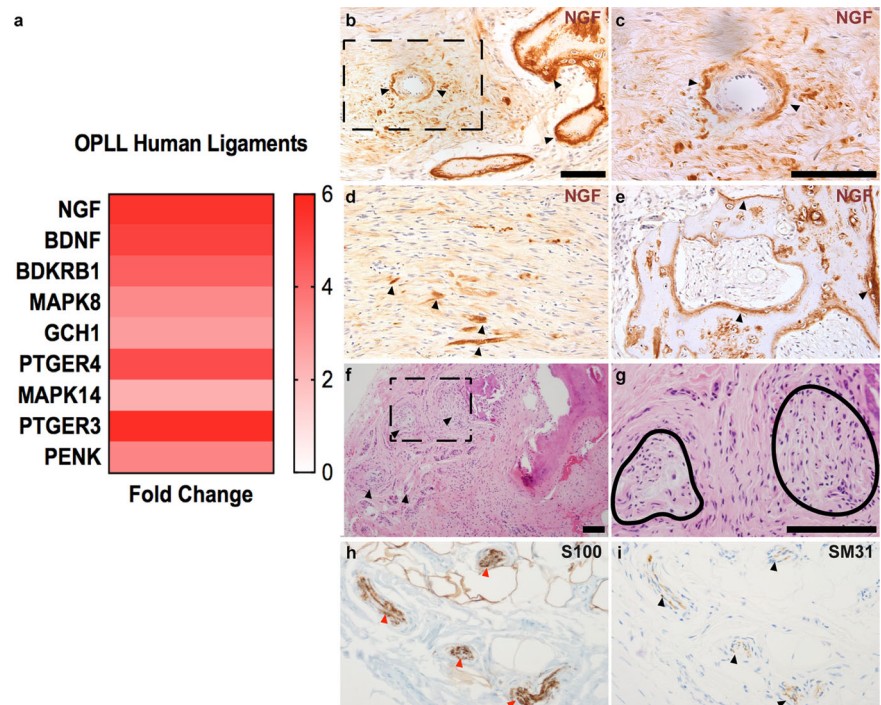

**Fig. 9 NGF-TrkA signaling and innervation within human heterotopic ossification (HO). a** NGF signaling among ligamentous cells derived from patients with ossification of the posterior longitudinal ligament (OPLL) [GEO microarray data (GSE5464)]. *NGF* expression was increased within the OPLL group relative to the control group (log fold change 5.48, $p = 0.02$). **b–e** NGF immunohistochemical staining within clinical examples of (**b**, **c**) perivascular cell-associated NGF at low and high magnification. Dashed black box shows the high magnification images in panel (**c**). **d**, **e** Immunostaining of NGF in early- and late-stage human HO, respectively. NGF immunostaining appears brown, while hematoxylin counterstain appears blue. Arrowheads indicate NGF immunostaining. **f–i** Axonal invasion within clinical examples of late-stage human HO. **f**, **g** Representative HO sites at low and high magnification, respectively. Arrowheads and black lines indicate HO-associated nerve fibers. **h**, **i** Immunohistochemical staining to highlight nerve fibers in human HO, including (**h**) S100 (red arrowheads) and (**i**) SM31 (black arrowheads). Scale bars: 100 µm. Statistical analysis was performed using a two-way Student's *t* test. $N = 12$ Human samples. See Supplementary Table S4 for a description of locations and demographics of human cases.

nerve terminals induce stem cell differentiation within the tooth[61]. A similar paradigm of neural-derived Hh signaling was considered in our model; however, we were not able to demonstrate levels of Hh receptors within mesenchymal cells of our dataset. Similarly, in our prior observations, Wnt signaling activation lies downstream of TrkA-expressing neuron sensitization to mechanical loads within the long bones[62]. While increased canonical Wnt signaling has been described as a molecular contributor to HO formation[60], in our dataset significant changes in Wnt signaling activation were not observed in the context of neurectomy. Downstream mediators of

HO formation released by local axons remains a subject of further investigation.

An important additional consideration is how neuro-immune crosstalk influences HO formation. A large body of literature suggests that acute inflammation is an early microenvironmental niche factor required for HO[53,63–66] (reviewed in ref. [11]). In our study, slight changes in both neutrophil and monocyte/macrophage gene signatures were observed in the presence of spared versus transected proximal nerves. For example, a modest shift in macrophage phenotype was observed with M1 to M2 phenotype upon denervation.

Flow cytometry showed overall similar numbers of myeloid cells, but a shift from Ly6c$^{low}$ to Ly6c$^{high}$ cells. In several experimental HO models, the positive regulatory role of macrophages on disease progression has been observed[65,67–70]. Likewise, we recently reported a prominent regulatory role of monocytes in our model of tenotomy-associated HO[44]. The extent to which NGF-TrkA signaling directly modulates immune cell migration or function in our model is not known. Although many inflammatory cells, including neutrophils[71], macrophages[72], and lymphocytes[73], have been reported to express TrkA, our results did not find appreciable Ngfr or Ntrk1 transcripts among inflammatory cell clusters. However, the role of inflammation as a prerequisite niche factor for HO is well established, and we cannot exclude that subtle alterations in the acute immune response may play a contributory role in our findings.

Finally, our study agrees with a large body of literature supporting the importance of neurovascular crosstalk in tissue repair. Nerves and blood vessels are well established to influence the growth and trajectory of one another as they grow to target tissues during organogenesis or repair[74,75]. Indeed, our own observations of a perivascular source for NGF are one example of this neurovascular crosstalk. In our own prior observations during long bone morphogenesis[23] in stress fracture[24], TrkA inhibition resulted in impaired vascular assembly. At least in the native skeleton, there is a clear positive regulatory relationship between invading axons and vascular ingrowth[23,24]. Our findings with Ngf deletion within the HO model support this functional coupling of vessels and nerves. Clinical observations with recently developed neutralizing antibodies to NGF have shown a high incidence of osteonecrosis and rapidly progressive osteoarthrosis[76,77], which may be another reflection of impaired angiogenesis in the context of neurotrophin antagonism. Indeed, in combination, our present and published work suggest a functional coupling of axons, blood vessels, and bone-forming mesenchyme, and that NGF-TrkA signaling is a primary mechanism that links these dependent cellular processes.

The present study has several additional limitations. First, neurectomy resulted in a prominent reduction in bone formation after injury, yet persistent cartilage was identified among neurectomized animals. Given the well-established phasic nature of HO in the burn/tenotomy model[12,13], we interpreted these results as a prominent delay in HO formation with neurectomy. Nevertheless, we have not entirely excluded the possibility that denervation completely abrogates the formation of bone independent of cartilage formation. Second, while our studies focused on the high-affinity receptor TrkA, we noted that p75 (Ngfr) expression was present locally within the injury site, particularly in pericytes/vascular SMCs. This raises interesting questions as to whether NGF-Ngfr signaling mediates important functions in HO, such as injury site vasculogenesis. A role for NGF-Ngfr signaling in VEGF-mediated angiogenic response has been observed in other contexts[78]. The potential role of NGF-p75 signaling in traumatic HO is an interesting avenue for future study. Third, our study focused on the potential molecular mediators by which local peripheral nerves regulate HO. Nevertheless, several studies have implicated Schwann cells as bone precursor cells in select contexts such as digit tip regeneration[30], and healing of the mandible[79]. The role of peripheral nerves as a direct cellular contributor to HO requires further use of specific transgenic animal models.

Several clinical features indicate the relevance of our findings to human disease. First, HO often occurs within proximity to large peripheral nerve fibers, especially after their injury, such as the ulnar nerve after fracture[80,81] or sciatic nerve after hip arthroplasty[82]. Second, the pain almost always precedes radiographic evidence of HO. Further, patients with HO almost universally suffer from chronic pain. Our experimental Ngf deletion studies demonstrate a potential targetable signaling pathway for clinical translation. Neutralizing antibodies for NGF are in the late stages of clinical development for osteoarthritic pain, including tanezumab and fasinumab, and may have potential off-label use for the prevention of HO[76,77]. Our aggregate data suggest that inhibition of NGF-TrkA signaling may have dual utility in HO patients, both as an analgesic and negative regulator of HO disease progression. Future studies must clarify the safety of antagonism of NGF signaling, particularly in light of our findings demonstrating impaired vascular assembly in the context of Ngf gene deletion.

## Methods

**Animal use.** All animal procedures complied with relevant ethical regulations for animal testing and research, and were carried out in accordance with the guidelines provided in the Guide for the Use and Care of Laboratory Animals from the Institute for Laboratory Animal Research (ILAR, 2011) and were approved by the Institutional Animal Care and Use Committee (IACUC) of the University of Michigan (PRO0007390) or Johns Hopkins University (MO16M226 and MO19M366). All animals were housed in IACUC-supervised facilities at 18–22 °C, 50% (±20%) of relative humidity, and 12-h light–dark cycle with ad libitum access to food and water, unless otherwise stated. Wild-type C57BL/6J mice were purchased from Jackson Laboratories (Bar Harbor, ME). NGF-eGFP mice were donated by the Kawaja laboratory, which express eGFP under the control of the mouse NGF promoter[37]. Mice with floxed NGF alleles were generated in the Minichiello laboratory[83]. TrkA$^{F592A}$ mice were donated from the Ginty laboratory, which are homozygous for a phenylalanine-to-alanine point mutation in exon 12 of the mouse Ntrk1 gene (F592A)[38]. This point mutation in TrkA$^{F592A}$ mice renders the endogenous TrkA kinase sensitive to inhibition by the membrane-permeable small-molecule 1NMPP1[38]. Mixed-gender, 6–8-week-old animals were used for all experiments. Wherever feasible, littermate analysis was performed while blinded to genotype. See Supplementary Table S2 for a list of transgenic lines used.

**Surgical procedures.** Trauma-induced HO was achieved via complete transection of the Achilles tendon with concomitant 30% total body surface area partial thickness burn of the dorsum, which induces endochondral ossification of the tenotomy site over a 9-week period, as per our prior reports[34–36].

In select experiments, surgical denervation was performed at the same time as HO induction. Proximal sciatic neurectomy was performed via a lateral thigh incision. Blunt dissection was performed through the biceps femoris to expose the sciatic nerve. The sciatic nerve was transected proximal to the trifurcation of the nerve with reflection and suturing of the proximal nerve stump to the nearby semitendinosus muscle to prevent reinnervation. As a control, sham nerve injuries consisted of surgical exposure of the sciatic nerve without transection. Incisions were sutured closed with 5-0 Vicryl suture.

In order to achieve Ngf deletion within the HO site, an adenovirus encoding either Cre recombinase (Ad-Cre, 1045-HT, Vector Biosystems, Malvern, PA) or GFP (Ad-GFP, 1060-HT, Vector Biosystems) was injected percutaneously in mice with Ngf floxed alleles[83]. The viral backbone of the adenovirus is human adenovirus type 5 with a cytomegalovirus promoter. A total of $1 \times 10^9$ pore-forming unit particles were diluted in 50 μl normal saline and injected both 48 h prior to surgery and 1 h post surgery. In select experiments, the spatial distribution of recombination was assessed using the same protocol of Ad-Cre injection in mT/mG animals[84].

In order to achieve temporal inhibition of TrkA catalytic activity in TrkA$^{F592A}$ animals, the small-molecule 1NMPP1 was used[85] (Aurora Analytics, LLC, Baltimore, MD). Purity (99.2%) was confirmed by HPLC-UV254, and characterization by proton nuclear magnetic resonance (400 MHz, dimethyl sulfoxide-$d6$ (DMSO-$d6$)) was consistent with structure. The stock solution was prepared at 200 mM by dissolving 1NMPP1 in DMSO. For 1NMPP1 administration, intraperitoneal injections were performed 24 h before, 2 h before and 24 h after injury using a 5 mM solution at a dosage of 17 μg/g body weight. In all cases, DMSO-containing vehicle was used for the control treatment. Animals were thereafter maintained on 1NMPP1 containing drinking water (40 μM 1NMPP1 in ddH$_2$O with 1% phosphate-buffered saline (PBS)-Tween-20).

In order to achieve systemic inhibition of TrkA in C57BL/6J animals, the TrkA inhibitor AR786 was used[39,40] (Array Biopharma, Boulder, CO, USA). The stock solution was prepared at 750 mg/ml of AR786 in DMSO) and resuspended in corn oil to make a mixture of 2% DMSO/corn oil. For AR786 administration, oral gavage was performed and mice were given 100 μl of the mixture at a dosage of 60 mg/kg daily for 3 weeks. In all cases, 100 μl of 2% DMSO/corn oil was used for the vehicle control treatment.

**RNA-seq-10× single-cell genomics.** The local injury site was microdissected at baseline (t0) and days 3, 7, and 21, and tissues were digested for 45 min in 0.3% Type 1 Collagenase and 0.4% Dispase II (Gibco) in Roswell Park Memorial Institute (RPMI) medium at 37 °C under constant agitation at 120 r.p.m. Digestions were subsequently quenched with 10% fetal bovine serum (FBS) RPMI and

filtered through 40 μm sterile strainers. Cells were then washed in PBS with 0.04% BSA, counted and resuspended at a concentration of ~1000 cells/μl. Cell viability was assessed with Trypan blue exclusion on a Countess II (Thermo Fisher Scientific) automated counter and only samples with >85% viability were processed for further sequencing.

Single-cell 3′ library generation was performed on the 10× Genomics Chromium Controller following the manufacturer's protocol for the v2 reagent kit (10X Genomics, Pleasanton, CA, USA). Cell suspensions were loaded onto a Chromium Single-Cell A Chip along with reverse transcription master mix and single-cell 3′ gel beads, aiming for 2000–6000 cells per channel. In this experiment, 8700 cells were encapsulated into emulsion droplets at a concentration of 700–1200 cells/μl, which targets 5000 single cells with an expected multiplet rate of 3.9%. Following generation of single-cell gel bead-in-emulsions (GEMs), reverse transcription was performed and the resulting Post GEM-RT product was cleaned up using DynaBeads MyOne Silane beads (Thermo Fisher Scientific, Waltham, MA, USA). The complementary DNA (cDNA) was amplified, SPRIselect (Beckman Coulter, Brea, CA, USA) cleaned, and quantified, and then enzymatically fragmented and size selected using SPRIselect beads to optimize the cDNA amplicon size prior to library construction. An additional round of double-sided SPRI bead cleanup is performed after end repair and A-tailing. Another single-sided cleanup is done after adapter ligation. Indexes were added during PCR amplification and a final double-sided SPRI cleanup was performed. Libraries were quantified by Kapa qPCR for Illumina Adapters (Roche) and size was determined by Agilent tapestation D1000 tapes. Read 1 primer, read 2 primer, P5, P7, and sample indices (SIs) were incorporated per standard GEM generation and library construction via end repair, A-tailing, adaptor ligation, and PCR. Libraries were generated with unique SIs for each sample. Libraries were sequenced on a HiSeq 4000 (Illumina, San Diego, CA, USA) using a HiSeq 4000 PE Cluster Kit (PN PE-410-1001) with HiSeq 4000 SBS Kit (100 cycles, PN FC-410-1002) reagents, loaded at 200 pM following Illumina's denaturing, and dilution recommendations. The run configuration was $26 \times 8 \times 98$ cycles for Read 1, Index and Read 2, respectively. Cell Ranger Single Cell Software Suite 1.3 was used to perform sample demultiplexing, barcode processing, and single-cell gene counting (alignment, barcoding, and a unique molecular identifier (UMI) count) at the University of Michigan Biomedical Core Facilities DNA Sequencing Core.

A total of ~4 billion reads were generated from the 10X Genomics sequencing analysis for 15 replicates (four replicates for days 0, 7, and 21; three replicates for day 3). The sequencing data were first preprocessed using the 10X Genomics Software Cell Ranger (10X Genomics Inc., Pleasanton, CA, USA) and aligned to mm10 genome. Downstream analysis steps were performed using Seurat. Each replicate was initially considered independently. Cells with <500 genes, with >60,000 UMIs, or expressing a fraction of mitochondrial UMIs >0.1, were filtered for quality control. Samples from each time point were grouped in four time-point sets. Genes present in <3 cells per set were discarded. For each time-point set, the downstream analysis included extraction of highly variable genes, normalization, and scaling (regressing against the number of gene per cell, number of UMI per cell, fraction of mitochondrial UMIs, and cell cycle genes). After quality control, the time-point sets were merged with canonical correlation analysis based on the intersection of the variable genes of the four time-point sets (3228, 5561, 12686, and 5136 cells for days 0, 3, 7, and 21, respectively). Unsupervised clustering (Louvain algorithm) was used to identify cell populations. To subcluster the pericyte/SMC cluster (1273 cells), the same procedure described above was followed after filtering out the cells not belonging to the cluster. The subclustering procedure extracted three clusters, labeled as pericytes (868 cells), SMCs (381 cells), and uncharacterized (24 cells). The uncharacterized cluster was discarded from the analysis. Repeating the analysis after discarding the uncharacterized cluster retrieved again the pericyte and SMC clusters.

Additional experiments performed for sciatic neurectomy and sham scRNA-seq samples ($N = 3$ samples per group, 8042 and 8781 cells for sham and neurectomy, respectively, obtained 7 days after injury) were next analyzed as described above. For analyses of mesenchymal subclusters (1559 and 1700 cells for sham and neurectomy, respectively), a proliferative cluster was identified, which could not be properly integrated through the use of regression of S- and G2M-phase scores and was removed prior to further analyses. To calculate cellular proliferation, the CellCycleScoring function of Seurat was used with the application of a more stringent filter of 0.1 in either the S.Score or G2M.Score to constitute a proliferating cell. For pathway analyses, KEGG pathways and manual annotation were used to generate gene lists of pathway activators. Average expression for each gene was calculated and the relative expression of cells following neurectomy versus sham was quantified.

**μCT imaging and analysis**. Mouse hindlimbs were harvested and imaged 9 weeks post injury (Bruker MicroCT, 35 μm resolution and 357 μA/70 kV/beam). Scans were analyzed by blinded operators manually splining around ectopic tissue and computing volumes at multiple thresholds (GE Healthcare v.2.2, Parallax Innovations rc18): unthresholded volume corresponding to total ectopic tissue in addition to 800 Hounsfield unit (HU).

**Histology and immunohistochemistry**. Specimens were harvested and placed in 4% paraformaldehyde at 4 °C for 24 h. After sequential washes in PBS × 3, samples

were decalcified in 14% EDTA (1:20 volume, Sigma-Aldrich) for 14–28 days at 4 °C. Sagittal sections of the Achilles tendon were obtained using cryosections at 10 or 50 μm thickness. Thin sections (10 μm) were used for all histochemical stains and some immunohistochemistry. Thick sections (50 μm) were used for immunohistochemical stains of nerve fibers. For cryosections, samples were cryoprotected in 30% sucrose overnight at 4 °C before embedding in OCT (Tissue-Tek 4583, Torrance, CA). Sagittal sections were mounted on adhesive slides (TruBond 380, Matsunami, Bellingham, WA). Histochemical stains included routine H&E, modified Goldner's trichrome, and Safranin O/Fast green. For immunohistochemistry, sections were washed in PBS washes in PBS × 3 for 10 min. When 50 μm sections were used, sections were next permeabilized with 0.5% Triton X for 30 min. Next, 5% normal goat serum was applied for 30 min and then incubated in primary antibodies overnight at 4 °C in a humidified chamber (see Supplementary Table S3). The following day, slides were washed in PBS, incubated in the appropriate secondary antibody, A594-Goat Anti-Rabbit IgG, or Goat Anti-Mouse IgG, for 1 h at 25 °C, and then mounted with 4′,6-diamidino-2-phenylindole mounting solution (Vectashield H-1500, Vector Laboratories, Burlingame, CA). Digital images of these sections were captured with ×10–100 objectives using upright fluorescent microscopy (Leica DM6, Leica Microsystems Inc., Buffalo Grove, IL) or confocal microscopy (Zeiss LSM780 FCS, Carl Zeiss Microscopy GmbH, Jena, Germany).

Twelve cases of nongenetic human HO were identified in our surgical pathology archives in the form of formalin-fixed, paraffin-embedded blocks (Johns Hopkins University), and the use of human pathology specimens complied with all relevant ethical regulations. All materials were coded so as to protect the confidentiality of personal health information and obtained under Johns Hopkins University Institutional Review Board approval with informed consent. All cases were reviewed by a bone pathologist to verify the diagnostic accuracy (A.W.J.). A summary of patient demographics is provided in Supplementary Table S4. Ten-micrometer-thick paraffin sections were prepared and pretreated with xylene and different concentrations of ethanol. Antigen retrieval was performed by using trypsin enzymatic antigen retrieval solution (Cat. No. ab970, Abcam) for 15 min at room temperatue (RT). After washing in PBS, the sections were incubated in 3% hydrogen peroxide for 20 min, followed by washing with PBS. Next, the slides were blocked with 5% goat serum (Cat. No. S-1000, Vector Laboratories)) for 30 min and then probed with 0.1% PBST diluted with the appropriate antibody overnight at 4 °C. The next day, the primary antibodies were washed off with 0.01% PBST and the slides were probed with 0.01% PBST diluted anti-rabbit IgG in 1:200 concentration for 1 h. After applying both VECTASTAIN® ABC Kit and DAB Peroxidase Substrate (Cat. No. SK-4100, Vector Laboratories), the samples were visualized. In some cases, dual immunohistochemistry was next performed. The slides were then rinsed thoroughly in PBS before reblocking with BIOXALL Endogenous Peroxidase and Alkaline Phosphatase Blocking Solution (Cat. No. SP-6000, Vector Laboratories) for 10 min and then 2.5% normal horse serum for 20 min at RT. After reblocking the slides, 0.1% PBST-diluted Mouse anti-PDGFRA primary was applied. After PBS wash, ImmPRESS™ AP Anti-Mouse IgG (Cat. No. MP-5402, Vector Laboratories) was applied for 30 min at RT and then VECTOR Red Alkaline Phosphatase Substrate Kit (Cat. No. SK-5100, Vector Laboratories) was applied for red color visualization.

Visualization of S100 was performed on a Benchmark Ultra Autostainer (Roche Diagnostics, Basel, Switzerland). Briefly, slides were deparaffinized and mild heat-induced epitope retrieval was achieved at 95 °C with CC1 solution (EDTA, pH 9; 950-224, Roche Diagnostics). A mouse monoclonal antibody against S100 (clone 4C4.9; predilute; 790-2914, Roche Diagnostics) was incubated for 4 min, followed by detection with a streptavidin–biotin iView Detection Kit (760-091, Roche Diagnostics). Visualization of SM31 was performed on a Leica Bond III (Leica Biosystems, Wetzlar, Germany). Briefly, slides were deparaffinized and incubated for 15 min with a mouse monoclonal antibody against Neurofilament-H (clone SMI-31, 1:20,000 dilution; NE1022, EMD Millipore, Burlington, MA), followed by BOND Polymer Refine Detection Kit (DS9800, Leica Biosystems).

**Microarray**. Microarray data from human spinal ligament cells were obtained from the publicly available Gene Expression Omnibus (GEO) dataset GSE5464. In this study, human spinal ligament cells from patients with OPLL and normal controls were grown in a deformable silicone chamber and subjected to cyclical stretching for 9 h[45]. All bioinformatics analyses were performed with R and Bioconductor packages: data were normalized and summarized using robust multiarray averaging with subsequent PCA visualization. Linear modeling of differential expression was performed using standard Bioconductor packages including *limma* package via R, as previously described[86–88]. All data underwent log 2 transformation, and *p* values were adjusted for the false discovery rate using Benjamini–Hochberg procedure.

**Western blotting**. Specimens were harvested, minced, and lysed in cold RIPA buffer (Thermo Fisher Scientific) with protease inhibitor (Cell Signaling Technologies, Danvers, MA). Proteins were separated by using a 4–20% gradient sodium dodecyl sulfate-polyacrylamide gel and transferred to a polyvinylidene difluoride (PVDF) membrane. Following the blotting, PVDF membranes were directly blocked with 5% nonfat milk in PBS with 0.1% Tween-20 (PBST) for 30 min at RT and then incubated with primary antibodies (Supplementary Table S3). Next, the blotted membrane underwent washing with PBST three times

and was incubated with a horseradish peroxidase-conjugated secondary antibody for 1 h at RT. The membrane was visualized with a ChemiDoc imaging system (Bio-Rad, Hercules, CA). Uncropped images are presented in Supplementary Fig. S16.

**ELISA**. The level of pro-NGF (Cusabio Biotech, Wuhan, China; Cat. No. CSB-EQ027721MO) in the uninjured tendon and injured Achilles tendon at days 3 and 7 post injury were measured following the instructions of the manufacturer. The optical density of each well was determined by using a microplate reader with absorbance set at 450 nm.

**von Frey testing**. Behavioral testing for von Frey filament stimulus and limb withdrawal were performed[89]. Mice were placed in a clear plastic confinement with a custom-manufactured metal mesh platform allowing full access to all paws. Mice were allowed to equilibrate to their surroundings for at least 15 min. The target for stimulation was mid-plantar left hind paw, with increasingly stiff von Frey filaments presented perpendicularly to the paw. Withdrawal of the hindlimb upon introduction or immediately upon the removal of the hair were considered positive responses. After multiple stimulations, a 50% response threshold was calculated[89].

**Open-field activity testing**. Video-based tracking (ANYmaze, Stoelting Co., USA) of open-field activity was measured in four adjacent cubicle plexiglass arenas (46 × 46 cm$^2$ floor area) for 10 min with a 1-min acclimation period prior to the start of the recording. Mice in their home cage were transported from the housing room to the procedure room. Each mouse was then identified by ear tag and placed into individual cages with bedding only and left undisturbed for 20 min. The floors of each quadrant were wiped with 70% ethanol and allowed to dry for 5 min. Four mice were then placed individually in one of the four arenas. With all four arenas occupied, the video-recording protocol was started. At the end of the protocol, the mice were returned to their home cage. The floors of the quadrants were cleared of debris, wiped with 70% ethanol, and allowed to dry for 5 min. The process was repeated until all the mice in the session were recorded. The open-field activity testing was repeated two times with 4 days between procedures.

**Flow cytometry**. After HO induction surgery, the soft tissue around the injury site was dissected from the posterior compartment between the muscular origin and calcaneal insertion of the Achilles tendon at the indicated time points. Tissue was digested for 45 min in 0.3% Type 1 Collagenase and 0.4% Dispase II (Gibco) in RPMI medium at 37 °C under constant agitation at 120 r.p.m. Digestions were subsequently quenched with 10% FBS RPMI and filtered through 40 μm sterile strainers. Specimens were blocked with anti-mouse CD16/32 and subsequently stained using the following antibodies: FITC:Ly6C, BV510:CD11b, APCH7:Ly6G, and BB700:F4/80 (see Supplementary Table S2). Stained and washed samples were run on a FACSAria II for cell sorting or LSRFortessa for analysis (BD) and analyzed using FlowJo software. See Supplementary Fig. S15 for further flow cytometry gating strategy information.

**Histologic image analysis and histomorphometry**. All images for quantification were obtained either with upright fluorescent microscopy (Leica DM6, Leica Microsystems Inc.) or confocal microscopy (Zeiss LSM780 FCS, Carl Zeiss Microscopy GmbH) centering around the distal Achilles tenotomy site. TUBB3$^+$ nerve fibers, CGRP$^+$ nerve fibers, and TH$^+$ nerve fibers were quantified using either the magic wand tool of Photoshop CC, 2017 with a tolerance of 30 (Adobe, San Jose, CA) using five random ×40 microscopical fields per sample or three-dimensional volumetric analysis of Imaris software v.9.3 (Oxford Instruments, Belfast, UK) using eight serial fields per sample within the injured tissue. A standardized region centered on the distal tendon was used (1000 pixel width by 2100 pixel height), which included the distal tenotomy site and immediately surrounding tissues. Longitudinal sectional slides were stained with Safranin O-Fast Green to assess C.Ar and were stained with H&E and modified Goldner's trichrome to evaluate B.Ar. The analyses were conducted in the whole Achilles tendon with the OsteoMeasure histomorphometry analysis system (OsteoMetrics, Decatur, GA, USA). Briefly, the stained slide images were transferred from the microscope with a video camera to Osteomeasure software for assessment of C.Ar and B.Ar by manually drawing the total area of cartilaginous tissue and bone tissue, respectively. Quantification and data generation was completed with automated measurement by the Osteomeasure software.

**Reactome and DAVID gene list analysis**. Reactome is a free, open-source, and peer-reviewed pathway database detailing molecular events and hierarchical relationships with extensive cross-referencing of external databases including Ensembl, Gene Ontology, PubMed, ChEBI, UniProt, OMIM, and many others[90]. Molecular participants for NGF and TrkA binding for mus musculus were identified, visualized, and exported using the Reactome pathway analysis web viewer. The resultant gene list was submitted to DAVID (database for annotation, visualization, and integrated discovery) for functional classifications and annotations[91].

**Statistics**. Quantitative data are expressed as a mean ± 1 SD with individual datapoints shown, unless otherwise stated. *$P < 0.05$, **$P < 0.01$, and ***$P < 0.001$ were considered significant, and adjustments were not made for multiple comparisons. The number of samples is also indicated in figure legends. A Shapiro–Wilk test for normality was performed on all datasets. Homogeneity was confirmed by a comparison of variances test. Parametric data were analyzed using an appropriate two-sided Student's *t* test when two groups were being compared, or a two-sided one-way analysis of variance was used when more than two groups were compared, followed by a post hoc Tukey's test to compare two groups. Nonparametric data were analyzed with a Mann–Whitney *U* test when two groups were being compared or a Kruskal–Wallis one-way analysis when more than two groups were compared. Sample size calculations were performed for experiments presented in Figs. 2 and 5 were based on an anticipated effect size range of 2.66–3.75, using our previously published data in adult mice[24]. For this scenario, with three samples per group, a two-sample *t* test would provide 80% power to detect effect sizes of at least 2.0 assuming a two-sided 0.05 level of significance. For Figs. 6 and 7, the sample size was calculated based on an anticipated effect size of 1.75 based on our prior data in TrkA mice. For this scenario, with five samples per group, a two-sample *t* test would provide 80% power to detect effect sizes of at least 1.5 assuming a two-sided 0.05 level of significance. All sample size calculation was performed by using G*Power version 3.1.9.2 (Franz Faul, Universitat Kiel, Germany).

**Reporting summary**. Further information on research design is available in the Nature Research Reporting Summary linked to this article.

## Data availability

Sequencing data are available through NCBI Gene Expression Omnibus at accession numbers "GSE126060" and "GSE163446." Microarray data from human spinal ligament cells were obtained from the publicly available Gene Expression Omnibus (GEO) dataset "GSE5464." All other relevant data supporting the findings of this study are included within the article and its Supplementary Materials files or from the corresponding authors upon reasonable request. Source data are provided with this paper.

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

## Acknowledgements

We thank the JHU microscopy facility, University of Michigan Center for Molecular Imaging, Biomedical Research Core Facilities DNA Sequencing Core, Amanda Fair, Nathan Lawera, Talis Rehse, Zaid Khatib, Kaetlin Vasquez, Jeffrey Lisiecki MD, Mohamed Garada for their technical assistance. A.W.J. was funded by NIH/NIAMS (R01 AR070773), NIH/NIDCR (R21 DE027922), USAMRAA through the Peer-Reviewed Medical Research Program (W81XWH-18-1-0121, W81XWH-18-1-0336) and Broad Agency Announcement (W81XWH-18-10613), American Cancer Society (Research Scholar Grant, RSG-18-027-01-CSM), and the Maryland Stem Cell Research Foundation. B.L. funded by the NIH (1R01AR071379, R61 AR078072), International Fibrodysplasia Ossificans Progressiva Association Research Award, and American College of Surgeon Clowes Award. A.W.J. and B.L. are funded by NIH (R01 AR079171) and DoD (W81XWH-20-1-0795). M.S. supported by Plastic Surgery Foundation National Endowment Award. C.H. supported by Howard Hughes Medical Institute Medical Research Fellowship. D.M.S. supported by Plastic Surgery Foundation Resident Research Award. T.L.C. was supported by the NIH/NIAMS (R01 AR068934), NIH/NIDCR (R21 DE027922), and the VA (Merit Award and Senior Research Career Scientist Award). The content is solely the responsibility of the authors and does not necessarily represent the official views of the National Institute of Health or the Department of Defense.

## Author contributions

Conception and design: A.W.J., B.L., and T.L.C.; funding and final manuscript approval: A.W.J. and B.L.; acquisition, analysis, and interpretation of data: S.L., C.H., S.M., R.J.T., Q.Q., S.N., C.A.P., Y.S., D.M.S., M.S., C.A.K., N.D.V., C.A.M., Y.W., H.A.R., J.X., S.M., A.K.H., P.S.C., and S.W.P.K.; donation of materials: L.M.; manuscript preparation: S.L., C.H., A.W.J., and B.L.

## Competing interests

Research unrelated to the work presented herein was supported in the James laboratory by Musculoskeletal Transplant Foundation (MTF) Biologics and Novadip Biosciences. A.W.J. is a paid consultant for Novadip and Lifesprout. This arrangement has been reviewed and approved by Johns Hopkins University in accordance with its conflict of interest policies. The authors have declared that no further conflict of interest exists.
