## [Peer Review File · Nature Communications]

Reviewers' Comments:

Reviewer #1:

Remarks to the Author:

General comments;

This is an excellent and comprehensive manuscript that describes the role of NGF-TrkA signaling as an essential regulator of neural ingrowth and subsequent formation of heterotopic ossification following traumatic soft tissue injury. The experiments are well controlled, well powered, and exceptionally well documented. The magnitude of effects of each intervention, which included denervation, NGF knockout, or TrkA inhibition (chemical-genetic and pharmacologic) was substantial and highly significant. The potential mechanisms of action from the scRNAseq analysis are interesting and plausible. The authors should be applauded for this outstanding work, and for taking a set of complementary approaches that leaves no doubt that this signaling axis plays an important role in the pathobiology of soft tissue HO.

Major specific comments:

The ingrowth of neuronal fibers into the tenotomy injury sites is impressively increased within the body of the injured tendon at sites of future HO. In contrast the increased presence of CGRP+ and TH+ nerve fibers appears to occur in the peritenon and the outskirts of the tendon. Are there other neuronal subpopulations that could account for the infiltration of TUBB3+ neurons in the tendon proper, or are these differences in localization due to sampling?

Previous work describing a role of neural mediators including substance P in HO formation. As in the current study, fairly impressive results were seen with inhibition of substance P signaling in those studies. Was substance P elaborated by any of the populations detected in the single cell analysis? Similarly did any of the Neurokinin receptors come up in the ligand-receptor analyses? Did sciatic nerve transection reduce substance P expression and/or secretion at sites of injury? One might look in various immune cell compartments to see if this was impacted by denervation. If it was not impacted among the cells that could be analyzed by scRNAseq that would also be interesting and not exclude changes in the neuronal populations not analyzed.

While the authors partly exonerate Hh and Wnt signaling as contributors to the effect of denervation by this analysis, these are known players in HO and orthotopic bone development.

The authors note that a switch towards FGF signaling occurs in the setting of denervation, and speculate that this is potentially protective in the discussion, but there is no evidence presented of this protective effect, and nothing to suggest that this is anything but a secondary change as a result of impairing osteogenic recruitment mechanisms, or due to prolonged fibroproliferative or cartilaginous phases of injury repair.

The decreased ingrowth of nerve fibers and subsequent diminished HO in denervated hindlimbs was impressive. The denervated hindlimbs showed only cartilage but not ossified matrix at 9 weeks. Was this simply a delay in the process, or abrogation of mineralization entirely? This would presumably require longer duration endpoints. Even a few observations beyond 9 weeks would be interesting to include.

Did the reduction in ossified matrix in denervated animals result in improved function?

In single cell analyses and IHC, NGF expression is essentially confined to SM MHC+ and PDGFRA populations considered vascular SM and pericyte populations. Could any of these populations signify fibroadipogenic progenitor cell populations given the expression of PDGFRA?

Specific minor comments:

Page 3, 2nd paragraph.

"Following activation, neurotrophin-receptor complexes..."

Please define neurotrophin as NGF in its first appearance

Pages 23-24. The description of results in TrkA-F592A KI mice should make clear that they received the small molecule inhibitor when appropriate (all received the small molecule, but the text switches back and forth to referring to them as just KI mice, or KI mice receiving the molecule). A preferred method would be to use the approach of the Figure legends, and state at the outset of this section of the results that all TrkA WT and KI mice received the inhibitor.

Reviewer #2:

Remarks to the Author:

Lee et al., investigates the relation between aberrant nerve growth of sensory and sympathetic fibers and heterotopic ossification. Overall, the study is ambitious and carefully conducted and the results are compelling. I just have a few points that could be clarified/improved:

1. The authors state that "pain pathways and NGF-TrkA axon ingrowth regulates...". I am not sure I understand this statement. It may not necessarily be pain pathways, as sympathetic innervation might also be important and the authors have not resolved the contribution of sensory versus sympathetic to the observed phenotype.
2. It would be good if the authors included a broader view on heterotopic ossification in the discussion. As the authors note in introduction, digit tip regeneration involves not only signaling, but also deposition of cells from nerves which contributes to regeneration. Consistently, bone and cartilage in head and neck is formed from neural crest cells during development. Thus, nerve contribution to heterotopic ossification may not necessarily only involve signaling events from the nerve, as investigated in Figure 8, but could also involve nerve-derived cells.
3. In the denervation experiment of figure 2 it is unclear how the nerve transected animals can show a modest 21.2% delay in paw withdrawal. If the nerve is truly transected there should be no response at all. The animal cannot feel without sensory nerves.
4. Fig S3. It is difficult to understand this figure as it seems that the across time points is shown but there is no temporal expression shown. Instead it looks like an average expression in all data (time-points) has been merged. The authors may want to clarify this in the figure legend or alternatively separate the data from the different time points and show all of them. I would do the latter, since it is a supplementary figure and there is not space limit.
5. Regarding the scSeq and histological analysis using the NGF-GFP mouse strain: There seems to be a discrepancy between these analyses since histological analysis finds more cell types than scRNA-seq. scRNA-seq has a limitation in that you can only sequence the cells that are successfully dissociated. In addition, there is often an underrepresentation of cell-types which are more fragile/sensitive to dissociation. Could this explain the discrepancies?
6. It is not clarified which Ad-Cre construct is used (which promoter etc) and which serotype which is used. The authors do not check if the local injection leads to system-wide recombination or not. A general/global NGF deletion does not affect their conclusions, but in this case they cannot claim a regional/local deletion. Thus, the authors may either want to weaken the claim for a local deletion of NGF or provide controls substantiating the claim.
7. The authors should clarify what mT/mG animals are when first described.
8. Dendrites, if applied, only refers to sensory, but not sympathetic axons so might be incorrect since the relative importance has not been resolved.
9. Axonal infiltration. Do axons infiltrate? Not a commonly used term and I am not sure if it is correct. I would use axon innervation or axon invasion.
10. It is stated in abstract that "...Regulates abnormal osteochondral differentiation...". I would rephrase and state for example "...drives abnormal osteochondral differentiation..." Or "is a driver

of... ". This is not a very important issue, but regulation for me is something that controls normal processes while dysregulation causes aberrant events.

Reviewer #3:

Remarks to the Author:

This manuscript addresses the very important issue of heterotopic ossification (HO) that occurs in the context of soft tissue trauma, a process that can be considered repair gone awry as a result of aberrant differentiation of mesenchymal progenitor cells in injured tissue. Acute and chronic pain are consequences and are treated with difficulty using existing medications. The authors identify NGF and its receptors on the axons of sensory and sympathetic neurons as mediating both developmental events important for normal periosteal innervation and local vascular development and, following soft tissue injury, in the emergence of HO. The latter is addressed in a series of studies employing the injured Achilles tendon in a mouse model of HO to elucidate the role(s) played by innervation, local NGF gene expression, NGF receptors and changes in gene expression in mesenchymal derivatives as well as immune cells. The authors point to sensory and sympathetic innervation in response to local increases in NGF synthesis and NGF signaling through its TrkA receptors as important for HO and suggest that treatments targeting key elements in the NGF/TrkA signaling pathway as having potential therapeutic benefit. The studies undertaken are logical, thoughtfully carried out, comprehensive in most respects and well described. The findings will be of interest to a large audience, including those who explore neurotrophin biology as well as those who care for patients suffering soft tissue injury.

I have a number of comments and suggestions:

1) The authors state that "our study demonstrates the potential central paracrine relationships between pericytes, TrkA-expressing axons, and mesenchymal progenitor cells in aberrant mesenchymal stem cell differentiation after trauma".

The claim appears to be correct, but details are missing as to the details needed to understand the relationship. Thus, it is important to define a more detailed characterization of the changes in the environment induced by tissue injury and how these changes are reflected in the actions of NGF and its receptors. These questions arise: a) which local cell type(s) are responsible for expressing NGF and its receptors, b) to what extent are the levels of the NGF protein increased in injury sites, c) is the increase in NGF predicted by increased mRNA levels reflected in increases in the pro- versus the mature form of the protein (noting of course that while the former would not activate TrkA the latter would), d) whether or not there are measurable changes in NGF signaling through TrkA in the site of injury, as predicted by the studies in which NGF was reduced and in the TrkA inhibitor studies, e) whether NGF signaling also occurs through resident p75NTR receptors, f) in which cell type(s) NGF signaling occurs to mediate HO, and g) would reducing NGF signaling in the context of soft tissue injury compromise the vascular supply of tissues undergoing repair. I realize that I have asked the authors for a great deal, but I see their work as providing a uniquely valuable perspective on a biological question raised by many but adequately studied in the past by very few.

2) Figure 1 reports on the model itself and is quite convincing. It would be good for the authors to clearly indicate within what domain the increase in density of fibers is reported in this and subsequent figures as this was not always clear. In fact, the local increase in NGF is characterized by both an overall increase in density and that within the tendon, so both measure are relevant. Indeed, the increase in axon staining event suggests the expected chemotropic effect of NGF, a phenomenon that would point to the most potent source of NGF within the tissue. The presence of CGRP fibers is very hard to see in panels n and n'; quantitation in panel o apparently records changes both within and outside the tendon.

3) Figure 2 reports on denervation studies. I was struck at the continued presence of axons within the injury site even 9 weeks after neurectomy. The authors should comment as it seems unlikely that full axotomy would leave any fibers at this time point. If denervation was not complete, the effects may be even more dramatic than pointed to. In the same figure we learn for the first time

in the manuscript that the changes can be interpreted as a delay in the process of ossification. The data for this is that cartilage continues to form in the denervated lesion. Two questions arise. The first is whether the process of ossification is delayed versus interrupted. If the former, longer post injury observations would show ossification. Was longer intervals interrogated? The second is how to explain the delay. Perhaps the authors could speculate as to what denervation caused that could explain the delay. Presumably this must be in some way linked to the absence of axons and possibly to ligands that they deliver to the injury site needed to maintain the pace of HO. Do the data in Figure 8 help to explain this? A small point is that the box in panel ab is not easily seen.

4) While increases in NGF mRNA are seen following injury, and while the analysis of the increases reveals its enrichment in pericytes and vascular smooth muscle cells is , the extent to which these changes are translated into protein, and the principal NGF isoforms (i.e. pro versus mature) are not. Evidence for the amount and type of NGF would be important. In this light, also, I think the title for this section indicating that trauma induces NGF signaling is not justified. It would be, of course, if one could detect local changes in the activation of NGF receptors, a suggestion listed above.

5) Figure 4 points to a number of cell types that express the NGF reporter gene and as such considerably complements the data in Figure 3. Thus, while pericytes and vascular smooth muscle cells are markedly positive, also positive are fibroblasts, macrophages, chondrocytes, osteoblasts and osteocytes. Expression in each cell type may have relevance for the HO and each source may contribute to the role played by NGF.

6) In Figure 5 the authors were wise to include TUBB3 density as a reduction in NGF levels would be predicted to negatively impact expression of CGRP and TH in their respective neurons. As a small point, the boxes in panels q and r are not easily seen.

7) I was interested to see the data for NGF receptor expression and not surprised that little if any TrkA was present, pointing to this receptor was likely only on the innervating axons. In contrast, p75NTR was present, albeit at rather low levels, in several cell types. These data raise the obvious possibility that local NGF signaling through p75NTR may mediate events relevant to HO. Indeed this seems a likely posit and one that could involve local paracrine relationships key to the events that impact mesenchymal cell differentiation. The authors may wish to consider this potential source of signaling in their model and to discuss what cell types might be impacted by signaling downstream from NGF/p75NTR. I am not suggesting an entirely new set of studies but rather that the authors point to the potential utility of exploring a role for this signaling pathway in contributing to HO.

8) In Figure 6 I was surprised to see the virtually instantaneous loss of TUBB3 in axons in the mouse expressing mutant TrkA. Were such fibers present in the control – i.e. the mutant mice before treating with 1NMPP1? If so, one would clearly question the validity of its use for these studies. Again, the boxes in panels q and r are nearly invisible.

9) The argument is made from RNA seq data that with loss of neural input there is a shift in signaling from TGF β to FGF. Perhaps, but again I think this claim begs direct evidence for data to support such a change.

10) It is not clear to me to what extent the human data presented support in a significant way the mouse data. We see that immunostaining for NGF is present in spindle shaped cells but not necessarily pericytes and vascular smooth muscle and in bone-associated cells. These findings are certainly not a mimic of the study in the mouse and do not powerfully support it.

11) The authors mean to say axons rather than dendrites on page 38, third line, second paragraph.

12) Regarding the trial of Tanezumab in OA, the data are now complete and have been reviewed by the FDA. The Discussion can thus be updated to account for this and it would be good as well to refer to the adverse events involving joints that were registered. Finally, and relevant to human data, one wonders whether reducing NGF levels may have an impact on the vascular supply to soft tissues undergoing repair, a topic raised above.

Response to Reviewer Comments

Reviewer #1 (Remarks to the Author):

This is an excellent and comprehensive manuscript that describes the role of NGF-TrkA signaling as an essential regulator of neural ingrowth and subsequent formation of heterotopic ossification following traumatic soft tissue injury. The experiments are well controlled, well powered, and exceptionally well documented. The magnitude of effects of each intervention, which included denervation, NGF knockout, or TrkA inhibition (chemical-genetic and pharmacologic) was substantial and highly significant. The potential mechanisms of action from the scRNAseq analysis are interesting and plausible. The authors should be applauded for this outstanding work, and for taking a set of complementary approaches that leaves no doubt that this signaling axis plays an important role in the pathobiology of soft tissue HO.

Major specific comments:

The ingrowth of neuronal fibers into the tenotomy injury sites is impressively increased within the body of the injured tendon at sites of future HO. In contrast the increased presence of CGRP+ and TH+ nerve fibers appears to occur in the peritenon and the outskirts of the tendon. Are there other neuronal subpopulations that could account for the infiltration of TUBB3+ neurons in the tendon proper, or are these differences in localization due to sampling?

Thank you for the excellent questions. Overall, nerve fibers (including CGRP+ and TH+ nerves) superficially enter into the edge of the transected tendon body in most tissue sections. In the revision, we have presented new images for Figure 1n and q to better reflect this (p. 6). As well, methods of quantification have been made more explicitly clear, which included an area of the distal tendon transection site and surrounding tissue (p. 51).

Previous work describing a role of neural mediators including substance P in HO formation. As in the current study, fairly impressive results were seen with inhibition of substance P signaling in those studies. Was substance P elaborated by any of the populations detected in the single cell analysis? Similarly did any of the Neurokinin receptors come up in the ligand-receptor analyses? Did sciatic nerve transection reduce substance P expression and/or secretion at sites of injury? One might look in various immune cell compartments to see if this was impacted by denervation. If it was not impacted among the cells that could be analyzed by scRNAseq that would also be interesting and not exclude changes in the neuronal populations not analyzed.

Thank you for the excellent point. *Tac1* (Substance P) was present in several populations within the dorsal root ganglia, previously defined as peptidergic fibers. These cell bodies showed a log2 fold change of 6.67 (maximum likelihood estimate) for *Tac1* in comparison to other neuronal cells within the DRG ¹.

Analysis of *Tac1* at the HO site showed overall infrequent expression across local cell types. Violin plots for both *Tac1* and *Tacr1* are shown below (Response Letter Figure 1). *Tac1* transcripts were found in 10% of endothelial cells, 0.5% of mesenchymal cells, and less than 0.1% in the rest of the cell clusters. In the ligand-receptor analysis, we find *Tacr1* ranked 443rd in expression among 589 receptors within local mesenchymal cells.

Response Letter Figure 1: *Tac1* and *Tacr1* expression at the HO site. Violin plots to show *Tac1* and *Tacr1* gene expression among 10 cell clusters. Cells isolated from N=3-4 animals per group per timepoint.

As to the expression of *Tac1* with or without neurectomy, we investigated this issue further (Response Letter Figure 2). In this dataset there was also low *Tac1* expression across all cells. Infrequent expression was identified in two mesenchymal clusters (cluster 2 and 5), and a slight increase in expression after neurectomy was identified in cluster 5. Rare inflammatory cells (clusters 1 and 3) also showed *Tac1* expression. Overall, these new datapoints do not support a significant role for Substance P signaling in our model. In addition, this does not support a robust change in *Tac1* expression among non-neuronal cell types after neurectomy.

Response Letter Figure 2: *Tac1* expression at the HO site with or without proximal sciatic neurectomy. (a) UMAP projection and clustering of pooled HO-derived cells from each condition at d 7 post-injury. (b) UMAP project of *Tac1* gene expression. (c) Violin plots to demonstrate *Tac1* gene expression among 10 cell clusters. Left side each of violin plot indicates sham surgery conditions, while right side indicates neurectomy conditions. Cells isolated from N=3 animals per group.

While the authors partly exonerate Hh and Wnt signaling as contributors to the effect of denervation by this analysis, these are known players in HO and orthotopic bone development.

We agree. The well-established role of Hh and Wnt signaling in HO has been briefly discussed, and we concur that Hh and Wnt signaling have been only partially exonerated (p. 40).

The authors note that a switch towards FGF signaling occurs in the setting of denervation, and speculate that this is potentially protective in the discussion, but there is no evidence presented of this protective effect, and nothing to suggest that this is anything but a secondary change as a result of impairing osteogenic recruitment mechanisms, or due to prolonged fibroproliferative or cartilaginous phases of

injury repair.

As suggested, we have changed our interpretation of the results to reflect that the change in FGF signaling may reflect several possibilities, including a prolonged fibroproliferative or cartilaginous phase of HO (p. 40).

The decreased ingrowth of nerve fibers and subsequent diminished HO in denervated hindlimbs was impressive. The denervated hindlimbs showed only cartilage but not ossified matrix at 9 weeks. Was this simply a delay in the process, or abrogation of mineralization entirely? This would presumably require longer duration endpoints. Even a few observations beyond 9 weeks would be interesting to include.

Thank you for the excellent point. We found a slight increase in cartilage among neurectomized animals at the 9 week timepoint (Fig. 2v). From this delayed increase in cartilage formation, we anticipate that there would be a corresponding delay in bone formation. Nevertheless, we have not entirely excluded the possibility that denervation completely abrogates the formation of bone in our model. This point has been raised as a limitation of the study (p. 41).

Did the reduction in ossified matrix in denervated animals result in improved function?

Thank you for the suggestion. We have added a new Supplementary Fig. S2 (p. 61), to assess behavior with video tracking analysis after injury with or without denervation. Although movement of the effected limb was reduced, the overall movement of animals was not. Indeed, in some metrics the animals with neurectomy demonstrated slightly increased movement. Over the long term, peri-articular HO is well known to reduce mobility², and we anticipate that chemical means to reduce frank bone in HO would overall lead to better functional outcomes. Changes to the main text (p. 9) and methods (p. 50) have been made.

In single cell analyses and IHC, NGF expression is essentially confined to SM MHC+ and PDGFRA populations considered vascular SM and pericyte populations. Could any of these populations signify fibroadipogenic progenitor cell populations given the expression of PDGFRA?

Thank you for the query. Our analysis suggested that fibroadipogenic progenitor cell markers were present within the mesenchymal cluster rather than the pericyte / vascular SMC groups. For example, the mesenchymal cell cluster showed high expression of *Pdgfra*, and low expression of *Acta2* and *Pdgfrb*. In contrast, the pericyte / vascular SMC cluster expressed high levels of *Pdgfrb*, *Acta2*, and *Myh11*, and low expression of *Pdgfra*. This distinction has been illustrated by volcano plots below (Response Letter Figure 3).

Response Letter Figure 3: Distinct expression profiles of mesenchymal versus pericyte / vSMC clusters. Volcano plots demonstrate distinct expression of *Pdgfra*, *Pdgfrb*, *Acta2* and *Myh11* across cell clusters at the HO site. Cells isolated from N=3-4 animals per group per timepoint.

Specific minor comments:

Page 3, 2nd paragraph.

“Following activation, neurotrophin-receptor complexes...”

Please define neurotrophin as NGF in its first appearance

This has been changed (p. 3).

Pages 23-24. The description of results in TrkA-F592A KI mice should make clear that they received the small molecule inhibitor when appropriate (all received the small molecule, but the text switches back and forth to referring to them as just KI mice, or KI mice receiving the molecule). A preferred method would be to use the approach of the Figure legends, and state at the outset of this section of the results that all TrkA WT and KI mice received the inhibitor.

Thank you, this has been changed as suggested (p. 24-27).

Reviewer #2 (Remarks to the Author):

Lee et al., investigates the relation between aberrant nerve growth of sensory and sympathetic fibers and heterotopic ossification. Overall, the study is ambitious and carefully conducted and the results are compelling. I just have a few points that could be clarified/improved:

1. The authors state that “pain pathways and NGF-TrkA axon ingrowth regulates...”. I am not sure I understand this statement. It may not necessarily be pain pathways, as sympathetic innervation might also be important and the authors have not resolved the contribution of sensory versus sympathetic to the observed phenotype.

Thank you for the good point. The reference to pain pathways has been removed as suggested in abstract, introduction and discussion (p. 2, 4, 39).

2. It would be good if the authors included a broader view on heterotopic ossification in the discussion. As the authors note in introduction, digit tip regeneration involves not only signaling, but also deposition of cells from nerves which contributes to regeneration. Consistently, bone and cartilage in head and neck is formed from neural crest cells during development. Thus, nerve contribution to heterotopic ossification may not necessarily only involve signaling events from the nerve, as investigated in Figure 8, but could also involve nerve-derived cells.

According to your suggestion, we have added a further discussion of the possible direct contribution of nerve-associated cells to HO, such as schwann cells (p. 42).

3. In the denervation experiment of figure 2 it is unclear how the nerve transected animals can show a modest 21.2% delay in paw withdrawal. If the nerve is truly transected there should be no response at all. The animal cannot feel without sensory nerves.

We thank the reviewer for their insightful comment. Intuitively, the reviewer is completely correct that a pain response should not be elicited if the area that is being tested is completely denervated (i.e., it should be totally numb to pain). However, these animals were tested at one week following the initial denervation. Although Wallerian degeneration begins in the distal stump almost immediately following transection, this process is not completed until approximately 3 weeks in both mice and rats^{3,4}. At 1 week following denervation, there are still axons which can transmit signals to their end-organ targets, potentially leading to responding to a lower filament than at baseline levels. This could explain why the neurectomized mice responded to a lower filament than they did at baseline levels, but displayed a 21.2% delay in paw withdrawal as compared to the sham group. Interestingly, there are also clinical conditions where painful responses occur in areas that are completely denervated, such as that seen in Anesthesia Dolorosa (or “painful numbness”). This is a chronic pain condition arising from damage to the trigeminal nerve where patients can experience complete numbness as well as constant excruciating pain^{5,6}.

4. Fig S3. It is difficult to understand this figure as it seems that the across time points is shown but there is no temporal expression shown. Instead it looks like an average expression in all data (time-points) has

been merged. The authors may want to clarify this in the figure legend or alternatively separate the data from the different time points and show all of them. I would do the latter, since it is a supplementary figure and there is not space limit.

As suggested, we have changed this to show the data across all timepoints individually (Response Letter Figure 4 below, also shown as revised Supplementary Fig. S4).

Response Letter Figure 4: Neurotrophic receptor tyrosine kinase 1 (*Ntrk1*) and Nerve growth factor receptor (*Ngfr*) expression within trauma-induced heterotopic ossification.

Cells at the injury site were collected prior to injury (day 0) and at 3, 7 and 21 days after HO induction. **(a)** *Ntrk1* expression within t-SNE plots, shown across each timepoint individually. **(b)** *Ntrk1* expression within violin plots of 9 cell clusters, merged across all timepoints. **(c)** *Ngfr* expression within t-SNE plots, shown across each timepoint individually. **(d)** *Ngfr* expression within violin plots of 9 cell clusters, merged across all timepoints. Cells isolated from N=3-4 animals per timepoint.

5. Regarding the scSeq and histological analysis using the NGF-GFP mouse strain: There seems to be a discrepancy between these analyses since histological analysis finds more cell types than scRNA-seq. scRNA-seq has a limitation in that you can only sequence the cells that are successfully dissociated. In addition, there is often an underrepresentation of cell-types which are more fragile/sensitive to dissociation. Could this explain the discrepancies?

Thank you for the good point. We concur that there is an underrepresentation of some cell populations within the scRNA-Seq dataset, including frank osteoblasts / osteocytes. For example, osteoblast or osteocyte markers such as *Ocn* and *Dmp1* are rarely seen in this dataset. As you point out, this likely explains the discrepancy in findings between scRNA-seq and histologic analyses.

6. It is not clarified which Ad-Cre construct is used (which promoter etc) and which serotype which is

used. The authors do not check if the local injection leads to system-wide recombination or not. A general/global NGF deletion does not affect their conclusions, but in this case they cannot claim a regional/local deletion. Thus, the authors may either want to weaken the claim for a local deletion of NGF or provide controls substantiating the claim.

Thank you for the question. We used a human adenovirus type 5 with a CMV promoter. This information was added to the methods section (p. 44). Systemic distribution of virus was assessed by harvesting multiple tissues after local injection of Ad-GFP into the HO site. Results showed significant distribution of Ad-GFP, especially in the liver and lung, but also to a lesser degree within the ipsilateral dorsal root ganglia (L5 DRG), spleen, kidney, and heart (Response Letter Figure 5; added as Supplementary Figure S7). Given this, reference to regional/local deletion has been removed from the revised text.

Response Letter Figure 5: Systemic distribution of GFP reporter activity following injury site Ad-GFP injection. Immunofluorescent images demonstrate that GFP reporter activity was seen in multiple tissues, including the (a) ipsilateral L5 DRG, (b) spleen, (c) kidney, (d) liver, (e) lung, and (f) heart. Scale bar: 50 μ m.

7. The authors should clarify what mT/mG animals are when first described.

Membranous TdTomato / Membranous GFP reporter mice have been defined at first mention (p. 20), in which all cells express TdTomato until Cre recombination, at which time mTdTomo is no longer expressed and instead mGFP is expressed.

8. Dendrites, if applied, only refers to sensory, but not sympathetic axons so might be incorrect since the relative importance has not been resolved.

This has been changed to axons as suggested (p. 20).

9. Axonal infiltration. Do axons infiltrate? Not a commonly used term and I am not sure if it is correct. I would use axon innervation or axon invasion.

Infiltration has been changed to invasion as suggested throughout the revision.

10. It is stated in abstract that "...Regulates abnormal osteochondral differentiation...". I would rephrase and state for example "...drives abnormal osteochondral differentiation..." Or "is a driver of..." ". This is not a very important issue, but regulation for me is something that controls normal processes while

dysregulation causes aberrant events.

This has been changed as suggested to “drives abnormal osteochondral differentiation” (p. 2).

Reviewer #3 (Remarks to the Author):

This manuscript addresses the very important issue of heterotopic ossification (HO) that occurs in the context of soft tissue trauma, a process that can be considered repair gone awry as a result of aberrant differentiation of mesenchymal progenitor cells in injured tissue. Acute and chronic pain are consequences and are treated with difficulty using existing medications. The authors identify NGF and its receptors on the axons of sensory and sympathetic neurons as mediating both developmental events important for normal periosteal innervation and local vascular development and, following soft tissue injury, in the emergence of HO. The latter is addressed in a series of studies employing the injured Achilles tendon in a mouse model of HO to elucidate the role(s) played by innervation, local NGF gene expression, NGF receptors and changes in gene expression in mesenchymal derivatives as well as immune cells. The authors point to sensory and sympathetic innervation in response to local increases in NGF synthesis and NGF signaling through its TrkA receptors as important for HO and suggest that treatments targeting key elements in the NGF/TrkA signaling pathway as having potential therapeutic benefit. The studies undertaken are logical, thoughtfully carried out, comprehensive in most respects and well described. The findings will be of interest to a large audience, including those who explore neurotrophin biology as well as those who care for patients suffering soft tissue injury.

I have a number of comments and suggestions:

1) The authors state that “our study demonstrates the potential central paracrine relationships between pericytes, TrkA-expressing axons, and mesenchymal progenitor cells in aberrant mesenchymal stem cell differentiation after trauma”.

The claim appears to be correct, but details are missing as to the details needed to understand the relationship. Thus, it is important to define a more detailed characterization of the changes in the environment induced by tissue injury and how these changes are reflected in the actions of NGF and its receptors. These questions arise: a) which local cell type(s) are responsible for expressing NGF and its receptors, b) to what extent are the levels of the NGF protein increased in injury sites, c) is the increase in NGF predicted by increased mRNA levels reflected in increases in the pro- versus the mature form of the protein (noting of course that while the former would not activate TrkA the latter would), d) whether or not there are measurable changes in NGF signaling through TrkA in the site of injury, as predicted by the studies in which NGF was reduced and in the TrkA inhibitor studies, e) whether NGF signaling also occurs through resident p75NTR receptors, f) in which cell type(s) NGF signaling occurs to mediate HO, and g) would reducing NGF signaling in the context of soft tissue injury compromise the vascular supply of tissues undergoing repair. I realize that I have asked the authors for a great deal, but I see their work as providing a uniquely valuable perspective on a biological question raised by many but adequately studied in the past by very few.

Thank you for the excellent suggestions. Since original review, we have performed additional experiments which specifically interrogate NGF and its receptors in HO induction injury. Firstly, we have more clearly defined which cell types at the site of injury express *Ngf*, *Ntrk1* and *p75*. As we previously observed by single cell RNA Seq, cell clusters representing pericytes and vascular SMCs were the predominant sample source of *Ngf* transcripts (Fig. 3). In addition, and by histological analysis in NGF-eGFP reporter animals, both some chondrocytes and osteoblasts were also NGF⁺ (Fig. 4).

Turning our attention to NGF receptors, we revisited our scRNA-Seq analysis, and observed that *Ntrk1* is very lowly expressed or not expressed in any HO site cell cluster (Supplementary Fig. S4, Response Letter Fig. 4 above). Importantly, local axons which are known to robustly express TrkA are not captured using single cell analysis of the tenotomy site, as their cell bodies are within the DRG. This lack of TrkA expression among osteoprogenitor/mesenchymal cells is also supported by our prior examinations of TrkA-LacZ reporter mice in bone development ⁷ and repair ⁸. Next,

Ngfr (p75) expression was assayed within the HO site as suggested, and was predominantly confined to the pericyte / vSMC cluster (Supplementary Fig. S4, Response Letter Fig. 4 above).

Second, protein expression of NGF was evaluated at the injury site by immunohistochemistry, western blot and ELISA. Importantly, in the original description of the NGF-eGFP reporter mouse⁹, the robustness of eGFP expression within a given tissue was observed to highly correlate with both NGF and proNGF expression by western blot. Results of additional experiments are shown below, which demonstrate a high correlation between NGF-eGFP reporter activity and NGF immunohistochemical staining at 1 week post-injury (Response Letter Figure 6; new Supplementary Figure S5). This is also in agreement with others who have used NGF immunohistochemical detection after experimental tendon injury¹⁰. Western blot for NGF, as well as western blot and ELISA for pro-NGF demonstrate an increase in expression at day 7 post injury, respectively. Changes to the manuscript can be found on p. 16.

Response Letter Figure 6: Protein expression of NGF and pro-NGF at the injury site. (a) NGF-eGFP reporter activity at the tenotomy site following 1 week post-injury. **(b)** Immunohistochemistry for NGF within the same image. **(c)** Merged images of (a,b). **(d)** NGF and pro-NGF at the injury site at days 0, 3, and 7 by western blot. **(e)** Quantification of NGF protein expression. **(f)** Quantification of pro-NGF protein expression. **(g)** ELISA based

measurement of pro-NGF expression at the HO site, days 0, 3 and 7-post injury. Scale bar: 100 μ m. N=3 animals per group for all assays. † P<0.10, * P<0.05, ** P<0.01.

Third, expression of pTrkA was analyzed within the ipsilateral lumbar DRGs corresponding to the injured area of the Achilles tendon. Immunohistochemical staining at 2 days post-injury demonstrated a 7.52 fold increase in pTrkA immunostaining in comparison to uninjured control (Response Letter Figure 7; added as Supplementary Figure S6). Overall, this new data supports the original model that injury site derived NGF binds to and activates TrkA on axons which innervate this region.

Response Letter Figure 7: Protein expression of pTrkA at the DRGs following tenotomy. (a) Immunohistochemistry for pTrkA in the DRGs among uninjured and (b) injured mice at 2 days post-injury. (c) Quantification of (a,b). Scale bar: 100 μ m. Each dot represents an individual animal. N=4 animals per group. *** P<0.001.

Finally, you raise an excellent point that reduced NGF signaling may have secondary effects on vascular ingrowth. Additional staining for CD31⁺ vascular channels was assessed across *Ngf* floxed animals treated with Ad-Cre or Ad-GFP control. As shown below (Response Letter Figure 8), *Ngf* deletion significantly reduces vascular ingrowth within the injury site. This material has been incorporated into new Supplementary Fig. S8, and changes to the manuscript can be found on p. 20.

Response Letter Figure 8: Vascularity of the HO site with *Ngf* deletion. Vascularity was assessed among the injury site of *Ngf* floxed animals treated with Ad-GFP or Ad-Cre. (a,b) CD31 immunofluorescent staining of the distal tenotomy site, shown in sagittal cross-section. Dashed

white line indicates the tendon edge. (c) Quantification of (a,b). Scale bar: 100 μ m. Each dot represents an individual animal. N=5 animals per group. *** P<0.001.

2) Figure 1 reports on the model itself and is quite convincing. It would be good for the authors to clearly indicate within what domain the increase in density of fibers is reported in this and subsequent figures as this was not always clear. In fact, the local increase in NGF is characterized by both an overall increase in density and that within the tendon, so both measure are relevant. Indeed, the increase in axon staining event suggests the expected chemotropic effect of NGF, a phenomenon that would point to the most potent source of NGF within the tissue. The presence of CGRP fibers is very hard to see in panels n and n'; quantitation in panel o apparently records changes both within and outside the tendon.

Thank you for the suggestions. For quantification of all images, a standardized region around the injured distal tendon was used (1,000 pixels width by 2,100 pixels height). This includes both a portion of the tendon body at the distal tenotomy site as well as the immediately surrounding tissues. This information has been added to the revision (p. 51). Our findings suggest that across most tissue sections there is ingrowth of both CGRP+ and TH+ nerve fibers into the peri-tendinous areas, as well as superficially into the tendon body itself. So as to improve clarity, we have shown new images for Figure 1n and q to more accurately represent this (p. 6).

3) Figure 2 reports on denervation studies. I was struck at the continued presence of axons within the injury site even 9 weeks after neurectomy. The authors should comment as it seems unlikely that full axotomy would leave any fibers at this time point. If denervation was not complete, the effects may be even more dramatic than pointed to. In the same figure we learn for the first time in the manuscript that the changes can be interpreted as a delay in the process of ossification. The data for this is that cartilage continues to form in the denervated lesion. Two questions arise. The first is whether the process of ossification is delayed versus interrupted. If the former, longer post injury observations would show ossification. Was longer intervals interrogated? The second is how to explain the delay. Perhaps the authors could speculate as to what denervation caused that could explain the delay. Presumably this must be in some way linked to the absence of axons and possibly to ligands that they deliver to the injury site needed to maintain the pace of HO. Do the data in Figure 8 help to explain this? A small point is that the box in panel ab is not easily seen.

We thank the reviewer for this important comment. We agree that if the denervation was complete, then we would not expect the presence of axons within the HO injury site. There are two important points to address this. First, the number of TUBB3, CGRP, and TH fibers are not statistically significant between the Neurectomy and Uninjured group (i.e., only the Sham group is significantly different). Second, it is well established that sciatic denervation in rodents causes sprouting of the neighboring saphenous cutaneous nerve into the previous sciatic territory¹¹⁻¹³.

A delay in HO was strongly suggested based on the phasic nature of the disease process, which proceeds from a fibroproliferative to cartilaginous to bony phase^{14,15}. In our model, the peaks of each phase are observed at 1, 3 and 9 weeks, respectively. Denervation caused a prominent delay of the cartilage phase from 3 to 9 weeks. This delay in cartilage formation among denervated injury sites is strongly suggestive of a corresponding delay in bone formation should later timepoints have been interrogated. Nevertheless, we have not entirely excluded the possibility that denervation completely abrogates the formation of bone independent of cartilage formation. This interesting point has been raised in the discussion (p. 41). As you suggested, the overall premise is that the delay is related to the paucity of axons and by extension axon-derived ligands delivered to the injury site. This has been made more explicitly clear in the discussion section (p. 41). Finally, the box in panel figure 2aa has been corrected (p. 10).

4) While increases in NGF mRNA are seen following injury, and while the analysis of the increases reveals its enrichment in pericytes and vascular smooth muscle cells is , the extent to which these changes are translated into protein, and the principal NGF isoforms (i.e. pro versus mature) are not. Evidence for the amount and type of NGF would be important. In this light, also, I think the title for this section

indicating that trauma induces NGF signaling is not justified. It would be, of course, if one could detect local changes in the activation of NGF receptors, a suggestion listed above.

Thank you for the good point. As above, protein expression of NGF and proNGF were evaluated at the injury site by immunohistochemistry, western blot and ELISA. In addition, phosphorylation of TrkA within the corresponding DRG was also assessed. Changes to the manuscript can be found on p. 16, and new Supplementary Figures S5 and S6.

5) Figure 4 points to a number of cell types that express the NGF reporter gene and as such considerably complements the data in Figure 3. Thus, while pericytes and vascular smooth muscle cells are markedly positive, also positive are fibroblasts, macrophages, chondrocytes, osteoblasts and osteocytes. Expression in each cell type may have relevance for the HO and each source may contribute to the role played by NGF.

Thank you for the good point, and our interpretation of the data has been amended to make this clearer (p. 17).

6) In Figure 5 the authors were wise to include TUBB3 density as a reduction in NGF levels would be predicted to negatively impact expression of CGRP and TH in their respective neurons. As a small point, the boxes in panels q and r are not easily seen.

As suggested, the boxes in Figures 5q and 5r have been changed (p. 22).

7) I was interested to see the data for NGF receptor expression and not surprised that little if any TrkA was present, pointing to this receptor was likely only on the innervating axons. In contrast, p75NTR was present, albeit at rather low levels, in several cell types. These data raise the obvious possibility that local NGF signaling through p75NTR may mediate events relevant to HO. Indeed this seems a likely posit and one that could involve local paracrine relationships key to the events that impact mesenchymal cell differentiation. The authors may wish to consider this potential source of signaling in their model and to discuss what cell types might be impacted by signaling downstream from NGF/p75NTR. I am not suggesting an entirely new set of studies but rather that the authors point to the potential utility of exploring a role for this signaling pathway in contributing to HO.

Thank you for the excellent point. As discussed above, Ngfr (p75) expression was assayed within the HO site, and was predominantly confined to the pericyte / vSMC cluster (revised Supplementary Fig. S4). This raises an interesting issue of autocrine NGF-Ngfr signaling within vascular wall resident cells of the HO site. Although testing NGF-Ngfr signaling would be considerably outside the scope of the current work, discussion of this point has been added (p. 41).

8) In Figure 6 I was surprised to see the virtually instantaneous loss of TUBB3 in axons in the mouse expressing mutant TrkA. Were such fibers present in the control – i.e. the mutant mice before treating with 1NMPP1? If so, one would clearly question the validity of its use for these studies. Again, the boxes in panels q and r are nearly invisible.

Thank you for the opportunity to clarify. As previously noted in the methods section, the control ‘uninjured’ limbs were contralateral limbs which in this case came from a mouse exposed to systemic 1NMPP1 for 9 weeks. Prolonged treatment with 1NMPP1 results in an expected reduction in nerve fiber frequency within the tendon, which is more clearly shown below (Response Letter Figure 9, new Supplementary Figure S10, p. 69). As observed, TrkA^{F592A} animals naïve to 1NMPP1 demonstrate similar levels of innervation of the peritenon as compared to other animal strains. However, after exposure to 1NMPP1 for 9 weeks, a 72% reduction in innervation of the uninjured tendon is seen. In the revision, we have made more clear that control animals represent animals in which the contralateral limb was assessed (p. 24-27). Finally, boxes in panels q and r have been modified as suggested (p. 26).

Response Letter Figure 9: Loss of nerve density within the uninjured Achilles tendon following 1NMPP1 treatment in TrkA^{F592A} animals. Analysis of uninjured Achilles tendon among TrkA^{F592A} untreated with 1NMPP1 and treated with 1NMPP1 for 9 wks. **(a,b)** TUBB3⁺ immunofluorescent staining of the uninjured tendon among 1NMPP1-untreated TrkA^{F592A} and 1NMPP1-treated TrkA^{F592A} mice, as visualized using sagittal cross-sections. Dashed white lines indicate edges of Achilles tendon. **(c)** Quantified TUBB3⁺ nerve density among 1NMPP1-untreated TrkA^{F592A} and 1NMPP1-treated TrkA^{F592A} mice. Each dot represents a single animal; N=3-5 per group. Statistical analysis performed using a Student's *t*-test. ** $P < 0.01$.

9) The argument is made from RNA seq data that with loss of neural input there is a shift in signaling from TGF β to FGF. Perhaps, but again I think this claim begs direct evidence for data to support such a change.

To further investigate a potential shift in signaling from TGF β to FGF signaling, we have performed immunohistochemistry for pSmad2, FGF2 and pERK1/2, focusing on the standard distal tenotomy site among samples that underwent either proximal sciatic neurectomy or sham surgery. Samples were analyzed at 1 week post-operatively, to coincide with the prior scRNA-Seq dataset. Our results further support a shift from TGF β to FGF signaling activation with neurectomy, including a reduction in pSmad2 immunostaining, and an increase in FGF2 and pERK1/2 immunostaining. These new images were added into revised Figure 8 (p. 33), with changes to the text on p. 32 and 34. New data is replicated below as Response Letter Figure 10.

Response Letter Figure 10: Shifting TGF β to FGF signaling activation following neurectomy at 1 week post-injury. Images shown are at the distal tenotomy site, with white dashed lines indicate the edge of the Achilles tendon. High magnification images are shown below. **(i,j)** pSmad2 immunofluorescent staining, sagittal cross-sectional view. **(k,l)** FGF2 immunofluorescent staining. **(m,n)** pERK1/2 immunofluorescent staining. Scale bar: 100 μ m.

10) It is not clear to me to what extent the human data presented support in a significant way the mouse data. We see that immunostaining for NGF is present in spindle shaped cells but not necessarily pericytes and vascular smooth muscle and in bone-associated cells. These findings are certainly not a mimic of the study in the mouse and do not powerfully support it.

Thank you for the good point. We have returned to previously stained human slides and found that perivascular cells in human HO likewise demonstrate NGF immunoreactivity. This has been added to the revised Figure 9b,c (p. 37).

11) The authors mean to say axons rather than dendrites on page 38, third line, second paragraph. **This has been changed as suggested (p. 39).**

12) Regarding the trial of Tanezumab in OA, the data are now complete and have been reviewed by the FDA. The Discussion can thus be updated to account for this and it would be good as well to refer to the adverse events involving joints that were registered. Finally, and relevant to human data, one wonders whether reducing NGF levels may have an impact on the vascular supply to soft tissues undergoing repair, a topic raised above.

Thank you for the good point. We have followed with interest the clinical side effects of anti-NGF, including osteonecrosis, subchondral insufficiency fractures and rapidly progressive osteoarthritis (RPOA). Indeed, it is interesting to speculate that impaired vascularity (as we observed in our model with *Ngf* deletion), is a common etiologic fracture in both these skeletal incidents and the possible prevention of HO. A reference to safety considerations of anti-NGF has been added in the discussion (p. 41).

References

1. Usoskin, D., *et al.* Unbiased classification of sensory neuron types by large-scale single-cell RNA sequencing. *Nature Neuroscience* **18**, 145-153 (2015).
2. Jamil, F., Subbarao, J.V., Banaovac, K., El Masry, W.S. & Bergman, S.B. Management of immature heterotopic ossification (HO) of the hip. *Spinal cord* **40**, 388-395 (2002).
3. Martini, R., Fischer, S., López-Vales, R. & David, S. Interactions between Schwann cells and macrophages in injury and inherited demyelinating disease. *Glia* **56**, 1566-1577 (2008).
4. Rotshenker, S. Wallerian degeneration: the innate-immune response to traumatic nerve injury. *Journal of neuroinflammation* **8**, 109 (2011).
5. Elahi, F. & Ho, K.W. Anesthesia dolorosa of trigeminal nerve, a rare complication of acoustic neuroma surgery. *Case reports in neurological medicine* **2014**, 496794 (2014).
6. Elahi, F., Luke, W. & Elahi, F. Intractable Facial Pain and Numb Chin due to Metastatic Esophageal Adenocarcinoma. *Case reports in oncology* **7**, 828-832 (2014).
7. Tomlinson, R.E., *et al.* NGF-TrkA Signaling by Sensory Nerves Coordinates the Vascularization and Ossification of Developing Endochondral Bone. *Cell Rep* **16**, 2723-2735 (2016).
8. Li, Z., *et al.* Fracture repair requires TrkA signaling by skeletal sensory nerves. *J Clin Invest* **129**, 5137-5150 (2019).
9. Kawaja, M.D., *et al.* Nerve growth factor promoter activity revealed in mice expressing enhanced green fluorescent protein. *The Journal of comparative neurology* **519**, 2522-2545 (2011).
10. Ahmed, A.S., *et al.* Compromised Neurotrophic and Angiogenic Regenerative Capability during Tendon Healing in a Rat Model of Type-II Diabetes. *PloS one* **12**, e0170748 (2017).
11. Devor, M., Schonfeld, D., Seltzer, Z. & Wall, P.D. Two modes of cutaneous reinnervation following peripheral nerve injury. *The Journal of comparative neurology* **185**, 211-220 (1979).

12. Markus, H., Pomeranz, B. & Krushelnycky, D. Spread of saphenous somatotopic projection map in spinal cord and hypersensitivity of the foot after chronic sciatic denervation in adult rat. *Brain research* **296**, 27-39 (1984).
13. Molander, C., Kinnman, E. & Aldskogius, H. Expansion of spinal cord primary sensory afferent projection following combined sciatic nerve resection and saphenous nerve crush: a horseradish peroxidase study in the adult rat. *The Journal of comparative neurology* **276**, 436-441 (1988).
14. Ranganathan, K., *et al.* Heterotopic Ossification: Basic-Science Principles and Clinical Correlates. *The Journal of bone and joint surgery. American volume* **97**, 1101-1111 (2015).
15. Vanden Bossche, L. & Vanderstraeten, G. Heterotopic ossification: a review. *Journal of rehabilitation medicine* **37**, 129-136 (2005).

Reviewers' Comments:

Reviewer #1:

Remarks to the Author:

The authors have done an outstanding job of addressing the reviewer's concerns.

Specific points

The additional figure panels illustrating the reinnervation at sites of injury, the additional details in methodology, and the revisions to the text and discussion are helpful and much appreciated.

The lack of a striking change in substance P (Tac1) based on single cell analyses is pertinent negative data that should be included in the discussion with previous relevant studies cited (Tuzmen et al, J Orthop Res 2018; Kan et al, J Cell Biochem 2011; Harris et al, Connect Tissue Res 2015 in addition to Salisbury et al., J Cell Biochem 2011 already cited). It should also be acknowledged in the discussion that the single cell analysis of injured tissues used in this study would not be likely to capture the sensory neuron cell bodies or their transcripts and so the lack of a difference in Tac1 in the present analysis does not exclude its participation. It is less surprising that it is absent from the various stromal cell and mesenchymal cell populations that participate in the model.

In addition to showing that PDGFRalpha+ lineages are present in among mesenchymal populations as the authors have shown in Figure 3, could the authors comment within the manuscript on the likely presence of fibroadipogenic progenitors within this compartment, and as possible drivers of this process. Giving the many different lineages that have been implicated in HO, it is helpful to place the elegant single cell analyses of this work in the context of previous work.

Again, the authors should be commended on their responsiveness and attention to detail in the preparation and revision of this excellent work.

Reviewer #2:

Remarks to the Author:

The authors have adequately responded to the comments with one exception, point 3 related to Sup Fig. S1 and experiment of Fig. 2. It remains unclear to me how sensory nerves in the skin in an animal with a complete transection of the sciatic nerve including nerve deflection and suturing can still signal from the periphery to the spinal cord. Such axotomized animals should have lost all peripheral sensitivity. Thus, in previous recorded studies using this experimental paradigm animals display no response threshold to mechanical (or any other) stimuli at all. The authors explain a modest increase in Von Frey thresholds by the fact that Wallerian degeneration is not complete 1 week after transection. However, Wallerian degeneration is the distal degeneration of a nerve stump that is no longer in contact with their central target and hence, can no longer transduce an action potential. Therefore, activating a transected distal nerve does not lead to any behavioral response. I do not understand why the authors starts discussing Anesthesia Dolores in humans which is unrelated to this experimental paradigm. Its mechanisms are unknown but can for example be the cause of damage only to the low-threshold mechanoreceptors leaving pain fibers intact or caused by spontaneous ongoing activity of the proximal nerve stump (phantom pain). Another strangeness of this experiment is that mice have a Von Frey threshold around 1.5 gram while the authors record 4 grams. I can only reconcile their results with that the sciatic nerve was not transected at mid-thigh level. I feel that it might be important for the authors to resolve this issue since it is used as a control for successful axotomy, while in fact the results suggest that the surgery was not successful.

Reviewer #3:

Remarks to the Author:

The manuscript has been extensively modified in response to my comments and I am satisfied with the care taken to respond and the additional studies and conclusions derived.

But I return to the statement in the current version "Neutralizing antibodies for NGF are in the late stages of clinical development for 14 osteoarthritic pain, including Tanezumab and Fasinumab, and may have potential off-label use for the prevention of HO. Our aggregate data suggest that inhibition of NGF-TrkA signaling may have dual utility in HO patients, both as an analgesic and negative regulator of HO disease progression." My view is that this concept needs qualification because, while true as stated, there is concern, based on the work now included within the paper, that excessive sequestration of NGF released by pericytes and vSMCs in the injured joint could compromise the integrity of the vacular responses to injury. I would ask the authors to consider this qualification. I would be even more excited if they would assess joints removed from those OA patients treated with Tanezumab versus controls to assess the status of NGF levels and possible changes in the vacular-bone interface in such patients.

Response to Reviewer Comments

Reviewer #1 (Remarks to the Author):

The authors have done an outstanding job of addressing the reviewer's concerns.

Specific points

The additional figure panels illustrating the reinnervation at sites of injury, the additional details in methodology, and the revisions to the text and discussion are helpful and much appreciated.

The lack of a striking change in substance P (Tac1) based on single cell analyses is pertinent negative data that should be included in the discussion with previous relevant studies cited (Tuzmen et al, J Orthop Res 2018; Kan et al, J Cell Biochem 2011; Harris et al, Connect Tissue Res 2015 in addition to Salisbury et al., J Cell Biochem 2011 already cited). It should also be acknowledged in the discussion that the single cell analysis of injured tissues used in this study would not be likely to capture the sensory neuron cell bodies or their transcripts and so the lack of a difference in Tac1 in the present analysis does not exclude its participation. It is less surprising that it is absent from the various stromal cell and mesenchymal cell populations that participate in the model.

Thank you for the suggestions. Additional discussion of Tac1 and corresponding citations as suggested have been added in the discussion (p. 40).

In addition to showing that PDGFRalpha+ lineages are present in among mesenchymal populations as the authors have shown in Figure 3, could the authors comment within the manuscript on the likely presence of fibroadipogenic progenitors within this compartment, and as possible drivers of this process. Giving the many different lineages that have been implicated in HO, it is helpful to place the elegant single cell analyses of this work in the context of previous work.

Thank you for the very interesting point. Our understanding of the degree of overlap of PDGFRa+ cells within tendon injury sites and FAPs continues to evolve. Since this is not the primary subject of the submitted manuscript, we have opted not to include the data for publication. However, we have included this information below.

In preliminary experiments, we used a combination of *Pdgfra* and *Ly6a* expression to identify 'FAP-like' cells within our scRNAseq dataset (Fig. 1A). This cell cluster broadly localizes to the 'mesenchymal' or 'mesenchymal progenitor cell' clusters previously designated. Expectedly, these *Pdgfra+Ly6a+* cells did show enrichment for osteochondral gene markers at the HO site (such as *Runx2*, *Sox9*, and *Acan*, as shown by violin plots below, Fig. 1B).

Figure 1: Identification of FAP-like cells within the HO site after trauma. Cells isolated for scRNA-Seq at 0, 7 and 42 days after injury. (A) UMAP Projections identifying a cluster of *Pdgfra*+*Ly6a*+ ‘FAP-like’ mesenchymal progenitor cells (MPCs). (B) Violin plots to demonstrate enrichment in osteochondral genes such as *Runx2*, *Sox9* and *Aggrecan (Acan)* among *Pdgfra*+*Ly6a*+ cells. 7,509 total cells examined.

In preliminary lineage tracing data using *Pdgfra*-CreER;R26-TdTomato reporter animals, we have obtained data suggesting that the designation of FAPs may not be the most appropriate in this model system. Here, one week after HO induction we observed that *Pdgfra* reporter + cells are widely distributed in the injury site and do express some osteochondral proteins by immunofluorescent staining, such as SOX9, OPN, RUNX2 and OSX. However, we observed that new adipocytes within the injury site are conspicuously not reporter positive, as shown by mutually exclusive distributions of PLIN1 immunofluorescence and *Pdgfra* reporter activity. In summary, *Pdgfra*+ cells (which co-express *Ly6a* by scSeq) significantly contribute to tendon-associated HO. However, the designation of these cells as FAPs may not be the most accurate terminology. As mentioned, the present manuscript has a central focus on innervation and HO, and owing to this fact we have opted to not include this data in the resubmission.

**Pdgfra-CreER;R26-TdTomato
One Week Post-Burn/Tenotomy**

PDGFRa-CreER; TdTomato

Figure 2: Lineage-tracing of *Pdgfra*⁺ cells following burn/tenotomy injury. Samples harvested from injury hindlimb 7 days following burn/tenotomy injury. Consecutive sections subjected to immunostaining for SOX9, Osteopontin (OPN), RUNX2, Osterix (OSX), or Perilipin (PLIN1).

Again, the authors should be commended on their responsiveness and attention to detail in the preparation and revision of this excellent work.

Thank you very much for your time in reviewing our resubmission.

Reviewer #2 (Remarks to the Author):

The authors have adequately responded to the comments with one exception, point 3 related to Sup Fig. S1 and experiment of Fig. 2. It remains unclear to me how sensory nerves in the skin in an animal with a complete transection of the sciatic nerve including nerve deflection and suturing can still signal from the periphery to the spinal cord. Such axotomized animals should have lost all peripheral sensitivity. Thus, in previous recorded studies using this experimental paradigm animals display no response threshold to mechanical (or any other) stimuli at all. The authors explain a modest increase in Von Frey thresholds by the fact that Wallerian degeneration is not complete 1 week after transection. However, Wallerian degeneration is the distal degeneration of a nerve stump that is no longer in contact with their central target and hence, can no longer transduce an action potential. Therefore, activating a transected distal nerve does not lead to any behavioral response. I do not understand why the authors starts discussing Anesthesia Dolores in humans which is unrelated to this experimental paradigm. Its mechanisms are unknown but can for example be the cause of damage only to the low-threshold mechanoreceptors leaving pain fibers intact or caused by spontaneous ongoing activity of the proximal nerve stump (phantom pain). Another strangeness of this experiment is that mice have a Von Frey threshold around 1.5 gram while the authors record 4 grams. I can only reconcile their results with that the sciatic nerve was not transected at mid-thigh level. I feel that it might be important for the authors to resolve this issue since it is used as a control for successful axotomy, while in fact the results suggest that the surgery was not successful.

We appreciate the reviewer's question, and thank you for the opportunity to better clarify. In terms of the response to mechanical stimuli in the setting of sciatic neurectomy, our surgery was performed at the level of the sciatic nerve prior to its trifurcation (Figure 3A). This spares the

saphenous nerve, which is part of the femoral nerve plexus, and provides sensory innervation to the medial aspect of the paw (Figure 3B). It is well established in the literature that complete sciatic denervation in rodents causes sprouting of the neighboring saphenous cutaneous nerve into previous sciatic territory¹⁻⁷. von Frey testing was performed in the mid-plantar section of the hindlimb, and so some degree of mechanical sensitivity from the intact saphenous nerve would be expected in our model.

Fig. 3: Mouse and rat hindlimb neuroanatomy. Obtained from Decosterd & Wolf (2000)⁸. (A) Diagram of the sciatic and saphenous nerves, their terminal branches and their dorsal root origins. (B) Different zones of the dorsal and plantar surfaces of the paw innervated by the sciatic nerve branches (sural and tibial) and the saphenous nerve. Images are shown in rat, but essentially identical neuroanatomy is found in mice (see also Duraku et al., (2012)⁹).

Second, we agree with the reviewer that normal control mice usually have a von Frey threshold of approximately 1.5 grams. However, this is usually measured on the lateral aspect of the mouse hindlimb, which is typically more sensitive than the mid-plantar section of the hind paw that was done in this experiment. In our experience, the threshold for mid-plantar responses are higher than those taken from the lateral aspect of the hind paw. Given that the average difference is only two filaments higher in the progression of filaments, we do not believe that this is dramatically different from what is expected.

Changes to make more clear the level of transection before the trifurcation of the sciatic nerve, as well as sparing of the saphenous nerve, have been added to the results (p. 8) and methods (p. 43). Thank you for your time in reviewing our resubmission.

Reviewer #3 (Remarks to the Author):

The manuscript has been extensively modified in response to my comments and I am satisfied with the care taken to respond and the additional studies and conclusions derived.

But I return to the statement in the current version "Neutralizing antibodies for NGF are in the late stages of clinical development for 14 osteoarthritic pain, including Tanezumab and Fasinumab, and may have potential off-label use for the prevention of HO. Our aggregate data suggest that inhibition of NGF-TrkA signaling may have dual utility in HO patients, both as an analgesic and negative regulator of HO disease progression." My view is that this concept needs qualification because, while true as stated, there is concern, based on the work now included within the paper, that excessive sequestration of NGF released by pericytes and vSMCs in the injured joint could compromise the integrity of the vascular responses to injury. I would ask the authors to consider this qualification. I would be even more excited if they would access joints removed from those OA patients treated with Tanezumab versus controls to assess the status of NGF levels and possible changes in the vascular-bone interface in such patients.

Thank you for the good point. This statement has been amended as suggested to now read, "Our aggregate data suggest that inhibition of NGF-TrkA signaling may have dual utility in HO patients, both as an analgesic and negative regulator of HO disease progression. Future studies must clarify the safety of antagonism of NGF signaling, particularly in light of our findings demonstrating impaired vascular assembly in the context of *Ngf* gene deletion."

While the suggested studies in assessing vascular changes with Tanezumab in OA patients would represent compelling additions to our current knowledge, they are considerably outside the scope of the present work which focuses on mouse and human HO.

Revisions can be found on (p. 42). Thank you for your time in reviewing our resubmission.

Reference

1. Markus, H., Pomeranz, B. & Krushelnycky, D. Spread of saphenous somatotopic projection map in spinal cord and hypersensitivity of the foot after chronic sciatic denervation in adult rat. *Brain Res* **296**, 27-39 (1984).
2. Molander, C., Kinnman, E. & Aldskogius, H. Expansion of spinal cord primary sensory afferent projection following combined sciatic nerve resection and saphenous nerve crush: a horseradish peroxidase study in the adult rat. *J Comp Neurol* **276**, 436-441 (1988).
3. Devor, M., Schonfeld, D., Seltzer, Z. & Wall, P.D. Two modes of cutaneous reinnervation following peripheral nerve injury. *J Comp Neurol* **185**, 211-220 (1979).
4. Vallin, J.A. & Kingery, W.S. Adjacent neuropathic hyperalgesia in rats: a model for sympathetic independent pain. *Neurosci Lett* **133**, 241-244 (1991).
5. Kingery, W.S. & Vallin, J.A. The development of chronic mechanical hyperalgesia, autotomy and collateral sprouting following sciatic nerve section in rat. *Pain* **38**, 321-332 (1989).
6. Kinnman, E., Aldskogius, H., Johansson, O. & Wiesenfeld-Hallin, Z. Collateral reinnervation and expansive regenerative reinnervation by sensory axons into "foreign" denervated skin: an immunohistochemical study in the rat. *Exp Brain Res* **91**, 61-72 (1992).
7. Puigdellivol-Sanchez, A., Prats-Galino, A., Ruano-Gil, D. & Molander, C. Sciatic and femoral nerve sensory neurones occupy different regions of the L4 dorsal root ganglion in the adult rat. *Neurosci Lett* **251**, 169-172 (1998).
8. Decosterd, I. & Woolf, C.J. Spared nerve injury: an animal model of persistent peripheral neuropathic pain. *Pain* **87**, 149-158 (2000).

9. Duraku, L.S., *et al.* Spatiotemporal dynamics of re-innervation and hyperinnervation patterns by uninjured CGRP fibers in the rat foot sole epidermis after nerve injury. *Mol Pain* **8**, 61 (2012).

Reviewers' Comments:

Reviewer #1:

Remarks to the Author:

The authors have addressed my concerns thoroughly.

Reviewer #2:

Remarks to the Author:

The authors have clarified the remaining concern.